# Auditory chain reaction: Effects of sound pressure and particle motion on auditory structures in fishes

Tanja Schulz-Mirbach[1]*, Friedrich Ladich[2], Alberto Mittone[3¤a], Margie Olbinado[3¤b], Alberto Bravin[3], Isabelle P. Maiditsch[2], Roland R. Melzer[1,4], Petr Krysl[5], Martin Heß[1]

1 Department Biology II, Ludwig-Maximilians-University Munich, Planegg-Martinsried, Germany,
2 Department of Behavioral and Cognitive Biology, University of Vienna, Vienna, Austria, 3 European Synchrotron Radiation Facility, Grenoble, France, 4 Bavarian State Collection of Zoology, Munich, Germany, 5 Department of Structural Engineering, University of California San Diego, La Jolla, CA, United States of America

¤a Current address: Cells—ALBA Synchrotron light source, Barcelona, Spain
¤b Current address: Swiss Light Source, Paul Scherrer Institut, Villigen, Switzerland
* schulz-mirbach@biologie.uni-muenchen.de

**Data Availability Statement:** data are available through https://doi.org/10.25365/phaidra.131

**Funding:** This study received funding from: European Synchrotron Radiation Facility (project

## Abstract

Despite the diversity in fish auditory structures, it remains elusive how otolith morphology and swim bladder-inner ear (= otophysic) connections affect otolith motion and inner ear stimulation. A recent study visualized sound-induced otolith motion; but tank acoustics revealed a complex mixture of sound pressure and particle motion. To separate sound pressure and sound-induced particle motion, we constructed a transparent standing wave tube-like tank equipped with an inertial shaker at each end while using X-ray phase contrast imaging. Driving the shakers in phase resulted in maximised sound pressure at the tank centre, whereas particle motion was maximised when shakers were driven out of phase (180°). We studied the effects of two types of otophysic connections—i.e. the Weberian apparatus (*Carassius auratus*) and anterior swim bladder extensions contacting the inner ears (*Etroplus canarensis*)—on otolith motion when fish were subjected to a 200 Hz stimulus. Saccular otolith motion was more pronounced when the swim bladder walls oscillated under the maximised sound pressure condition. The otolith motion patterns mainly matched the orientation patterns of ciliary bundles on the sensory epithelia. Our setup enabled the characterization of the interplay between the auditory structures and provided first experimental evidence of how different types of otophysic connections affect otolith motion.

## Introduction

Otoliths of modern bony fishes (Teleostei) are solid calcium carbonate biomineralisates in the inner ear and play an important role in hearing [1–4]. Teleost otoliths show a tremendous diversity in morphology, especially with respect to shape (e.g. [5–8]). Despite this known morphological diversity, it remains elusive how different otolith shapes affect sound-induced otolith motion and contribute to hearing in fish [4,9,10]. Identifying factors that determine

no. LS-2752, TSM, MH, FL); https://www.esrf.eu/
Bavaria California Technology Center (BaCaTeC,
project no. [2017]-1, TSM, PK); http://www.
bacatec.de/en/ Austrian Science Fund (project no.
P31045, FL), https://www.fwf.ac.at/en/ Münchener
Universitätsgesellschaft (MUG, project no. 31983,
TSM, MH), https://www.unigesellschaft.de/
foerderprojekte.html. The funders had no role in
study design, data collection and analysis, decision
to publish, or preparation of the manuscript.

**Competing interests:** The authors have declared
that no competing interests exist.

otolith motion and thus inner ear stimulation is even more challenging in teleost taxa that
evolved ancillary auditory structures (for an overview see [11,12]). These structures are morphological modifications such as connections between the gas-filled swim bladder and the
inner ears (= otophysic connection), which are often correlated with a widened range of
detectable frequencies (especially at frequencies > 700 Hz) and increased auditory sensitivity
(e.g. [13–15]). If an otophysic connection is present, sound is thought to stimulate the inner
ears along two pathways, a direct and an indirect one (Fig 1A; [16]). The direct stimulation
pathway is independent of the presence of a swim bladder or any gas-filled bladder in the fish
[16,17]. Sound-induced particle motion results in a to and fro motion of the fish, whereas all
soft tissue is likely to move in phase with the surrounding water due to similarities in density
(e.g. tail muscle tissue in fish: 1.05 g/cm$^3$; [18]). As the otoliths are denser than the underlying
sensory epithelium, they act as inertial masses and lag behind the motion of the soft tissue,
thereby creating a relative motion [19,20]. This relative movement between otolith and sensory
epithelium deflects the ciliary bundles of the sensory hair cells and maximizes stimulation if
the bundle is pivoted in the direction of the kinocilium [21,22]. In the indirect stimulation
pathway, sound pressure fluctuations compress and decompress the gas in the swim bladder,
resulting in oscillating swim bladder walls [16]. The motion of swim bladder walls thereby creates local particle motion. This may stimulate the inner ears in a way similar to sound-induced
particle motion, especially when the distance between swim bladder and inner ears is short
due to the coupling of the two structures. The swim bladder can be directly coupled to the
inner ears via anterior swim bladder extensions such as present in several sciaenid species (e.g.
*Bairdiella chrysoura* [13], *Micropogonias undulatus* [23]), the holocentrid genus *Myripristis*
[24], or the cichlid genus *Etroplus* [25,26]. In all otophysans such as carp or goldfish, the swim
bladder is connected to the inner ears through the Weberian apparatus [27].

Only few experimental and theoretical studies (based on mathematical modelling) have
investigated the factors influencing otolith motion as well as the interplay between inner ear
components and ancillary auditory structures such as otophysic connections [10,19,20,28–31].
A recent experimental study successfully visualized sound-induced *in situ* motion of otoliths
in the orange chromide *Etroplus maculatus* (Bloch, 1795) (Cichlidae) using hard X-ray phase
contrast imaging [30]. However, the sound field in the Plexiglas® tank equipped with a small
underwater speaker (Daravoc) was complex and clearly deviated from values expected under
free field conditions ([30], see also [32]). We therefore aimed to improve the tank acoustics
and to develop a setup enabling experiments either under sound-induced particle motion or
under sound pressure condition.

To test the newly developed setup, we chose two species representing two types of otophysic
connections, i.e. goldfish, *Carassius auratus* (Linnaeus, 1758) (Otophysa, Cyprinidae) and the
Canara pearlspot, *Etroplus canarensis* Day, 1877 (Cichlidae). Goldfish is well investigated with
regard to inner ear and swim bladder morphology. For *Etroplus*, data on the gross morphology
of the auditory structures are available [15,25,26,33,34]. The goldfish is characterized by an
indirect swim bladder-inner ear connection via the Weberian apparatus. The Weberian apparatus is a chain of three ossicles (but see its variability in Siluriformes, [35,36]), namely tripus,
intercalarium, and scaphium plus the claustrum as well as ligaments that interconnect the ossicles [37,38]. In *E. canarensis*, the swim bladder contacts the inner ears through two anterior
swim bladder extensions [25]. In *E. maculatus*, each anterior swim bladder extension consists
of 1) a gas-filled part contacting the bone surrounding the lagena and 2) a tissue pad that
comes close to the posterior and horizontal semicircular canals and to a recessus posterior to
the utricle [26]. Goldfish and *E. maculatus* have improved auditory abilities in terms of the
detectable frequency range and relative auditory sensitivity [15,39] compared to species without swim bladders such as flatfish (*Pleuronectes platessa* [17]) or species that lack an otophysic

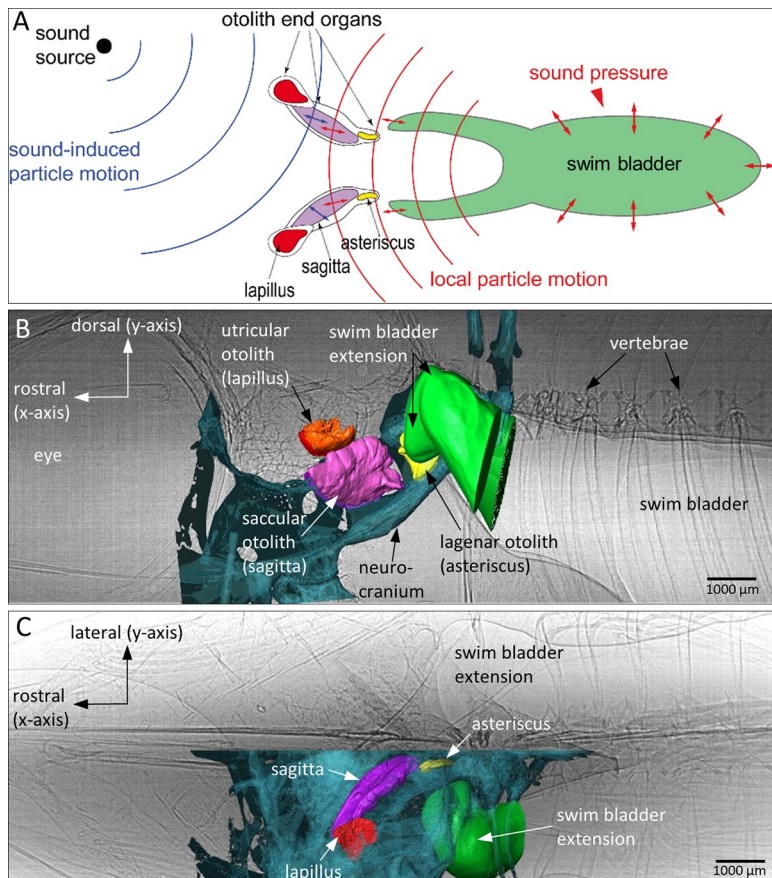

**Fig 1. Overview of an otophysic connection represented as a scheme and as found in the cichlid *Etroplus canarensis*.** (**A**) Hypothesized direct (blue) and indirect (red) pathways of inner ear stimulation in teleost fishes. All fish, whether or not they possess a gas-filled bladder, can detect sound via the direct stimulation pathway. If the swim bladder comes close to the inner ears or is coupled to the inner ears (otophysic connection), oscillations of the swim bladder walls provoked by sound pressure can be reradiated as local particle motion which stimulates the inner ears. 3D reconstructions of the otoliths, part of the bones, and the anterior swim bladder region are superimposed on the 2D radiographs of the same individual of *E. canarensis* (based on a pixel size of 6.1 μm, ID17) seen in (**B**) lateral and (**C**) ventral view. Scheme in (**A**) adapted and modified from Schulz-Mirbach et al., 2019 [4].

connection (Holocentridae: *Sargocentron*, no otophysic connection *vs. Myripristis*, otophysic connection present [12,14]). Moreover, the goldfish is considered a "gold standard" in fish bioacoustics because its auditory abilities have been intensively studied using different behavioural and electrophysiological methods [40–43].

Here, we address the functional morphological question of how different types of otophysic connections affect otolith motion. Two formerly stated hypotheses were tested. 1) In otophysans such as goldfish, the saccule is assumed to play an important role in perceiving sound pressure. The local particle motion generated by the swim bladder and transduced via the Weberian apparatus, the sinus impar, the sinus endolymphaticus, and the transverse canal to the saccules is expected to result in a motion of the saccular otoliths that is in direct line with the flow of the endolymphatic fluid [33,44]. This otolith motion might be supported by the three wings arranged along the rostrocaudal axis of the saccular otolith [44,45]. 2) The bipartite structure of the anterior swim bladder extensions in the cichlid genus *Etroplus* [25,26] may facilitate the transduction of sound pressure to the utricle [26]. The modified utricular otolith

and the macula utriculi in *Etroplus maculatus* have been interpreted in terms of a more important role of the utricle in audition [26,33].

## Material and methods

### Animals

Fishes (*Carassius auratus*, *Etroplus canarensis*) were purchased from French aquarium traders and kept at the animal care facility of the European Synchrotron Radiation Facility (ESRF, Grenoble, France). For each of the three measurement sessions in 2018 (June, July, September), the animals were maintained in two 20L-plastic aquaria (5–7 individuals per aquarium) under a 12:12 h light-dark cycle. Each aquarium was equipped with an internal filter and a bubble stone to aerate the water.

Prior to each sound experiment, the test individual was deeply anesthetized and euthanized with a neutral-buffered solution of 0.4% ethyl 3-aminobenzoate methansulfonate (MS-222, Sigma Aldrich, France). After opercular movements had ceased (within ca. 5 minutes), we waited another 10–15 minutes before further handling of the fish. Standard length and body weight of the fish were then measured to the nearest millimetre using a scale paper and to the nearest 0.1 g using a Scaltec SBC22. All experiments at the ESRF were conducted in accordance with ethical guidelines, i.e. article 3, point 1 of the EU directive 2010/63, accepted by all EU countries. As no living animals were used during the imaging procedure, no ethical approval was required.

### 2D radiography

In total, eight specimens per species were studied with goldfish possessing a mean standard length of 62 ± 2.2 mm (ID17) and 54 ± 1.6 mm (ID19) and a body weight of 7.5 ± 1.7 g (ID17) whereas *E. canarensis* had a mean standard length of 51 ± 1.0 mm (ID17) and 51 ± 2.7 mm (ID19) and a body weight of 6.0 ± 0.3 g (ID17).

For each experiment, the freshly euthanized fish was placed in the tank with its head located in the centre. Imaging was performed with the fish seen in lateral (Fig 1B), dorsal (Fig 1C), and frontal views; the latter orientation ("frontal view") was performed only in two specimens of *E. canarensis*. In dorsal and lateral views, sound impinged along the rostrocaudal axis of the fish, whereas, in frontal view, sound impinged on the body flanks. The fish rested on a piece of foam that was glued to a Plexiglas® holder. To provide as many degrees of freedom for the fish's body to move in the sound field, it was fixed to the foam with two or three insect pins that were pierced subcutaneously through the body flanks (lateral and frontal views) or through the caudal peduncle (dorsal view) and through the anterior-most portion of the upper and lower jaws (lateral, dorsal, and frontal views). To minimize detrimental effects of X-ray-induced tissue damage, experiments performed in the same orientation (e.g. fish seen in lateral view) and using the same species, but different individuals, alternately started with the in phase (0°) and the out of phase condition (180°; for an explanation of the two conditions see "setup design"). Hence, we started using the in phase condition and the highest studied sound pressure level (SPL; step 1, Table 1) in one specimen and using the out of phase condition at a lower SPL (e.g. step 4, Table 1) in another individual of the same species.

The motion of the structures was captured at an image acquisition rate of 98.99 frames per second (= fps, ID17) or 198.02 fps (ID19) when presenting the 200 Hz stimulus. The mismatch between stimulus frequency and image acquisition rate enabled sampling one oscillation (duration of 5 ms) of the 200 Hz pure tone at 50 (ID17) or 100 positions (ID19). In all experiments, the integration time was 1 ms; a pixel size of 6.1 μm (ID17) or 3.67 μm (ID19) was achieved. Images were recorded using a frame size of 2,560 x 1,100 pixels as 32-bit raw-files

**Table 1. Overview of sound pressure levels (SPLs) measured at the centre of the tank and the difference of SPLs and particle acceleration levels (PALs) between the in phase (0˚) and out of phase (180˚) condition.**

| | hydrophone (Brüel & Kjær 8103) | | | | p-a sensor (Lab Vienna) | |
| --- | --- | --- | --- | --- | --- | --- |
| | SPLs (dB re 1 μPa) | | | | SPL (0˚) -SPL (180˚) | PAL (0˚) -PAL (180˚) |
| | ID17 | | ID19 | | | |
| ANL | 117.5 | | 118.1 | | | |
| step | 0˚ | 180˚ | 0˚ | 180˚ | | |
| 1 | > 177.2 | 165.5 | > 177.2 | 157.3 | | |
| 2 | > 177.2 | 161.6 | 177.2 | 155.4 | | |
| 3 | 169.6 | 154.6 | 170.7 | 151.1 | +24.8 | -8.7 |
| 4 | 162.6 | 148.6 | 164.3 | 146.2 | +26.0 | -8.2 |

Measurements were performed using a miniature hydrophone (Brüel & Kjær 8103) and the p-a sensor. The difference of SPLs was calculated from SPL values given in dB re 1 μPa, that of PALs was obtained from PAL values given in dB re 1 μm/s$^2$. ANL, ambient (background) noise levels.

(ID17) or using 2,016 x 2,016 pixels as 16-bit tiff-files (ID19) with a photon energy of 70 keV (ID17: monochromatic beam, DeltaE/E approx. 10–4) or 59.2 keV (ID19), respectively. The detected photon flux was estimated at $3.5*10^7$ photons/mm$^{2*}$ms (ID17). At ID19, the field of view was not large enough to capture the region of interest in goldfish (anterior swim bladder portion, Weberian ossicles, otoliths) in one stack; thus, two overlapping stacks were produced for each condition (orientation & sound presentation).

At ID17, the detection system consisted of a PCO edge 5.5 coupled with 1x optics, yielding a final isotropic pixel size of 6.1 μm. At the entrance screen of the detection system, a 250 μm LuAG:Ce scintillator was used to convert the X-rays into visible light [46]. To achieve a sufficiently high photon flux density at ID19, the beamline was operated in the "pink beam" configuration, i.e. the white radiation from a wiggler insertion device was filtered only by Al, C and Cu attenuators as well as the mandatory Be window in the optical beam path. Accordingly, a narrow-bandwidth X-ray spectrum was available at the same position which was characterized by a high amount of photons and a homogeneous wave used by the propagation-based imaging (PBI) technique (see next paragraph). Another indirect system was used as a detector for fast micro-radioscopy: two photo-lenses in tandem-design (face-to-face, with an effective 3.0x magnification, Hasselblad) were used to project the luminescence image of a 200 μm LuAG:Ce single-crystal scintillator onto a (first generation) PCO DIMAX camera.

In conventional X-ray imaging, the contrast obtained is based on the absorption properties of the investigated object. In the case of soft tissue, the investigated materials are composed mainly by low Z elements; this limits the image contrast achievable using conventional absorption-based X-ray imaging. X-ray phase-contrast imaging (XPCI; [47]) helps overcome such limitation because it is based on detecting the phase shift induced by the presence of an object. Hence, high image contrast can also be achieved for soft tissue. The technique employed here is so-called propagation-based imaging (PBI; [48]), which is optimal whenever high spatial and temporal resolution is needed. PBI requires a highly spatially coherent beam and an optimized distance between the object and the detector in order to enhance the contrast between different interfaces thanks to Fresnel diffraction [48]. The characteristics of synchrotron radiation at ID17 and ID19 perfectly fulfil the requirements of this technique.

## Setup design: Standing wave tube-like tank

The tank construction followed the standing wave tube design described by Hawkins & MacLennan (1976) [17]. Nonetheless, imaging at the synchrotron radiation facility required several

modifications such as "X-ray-transparent" material (Plexiglas®) instead of steel, thin tank walls to minimize beam absorption, and a light tank weighing no more than 5 kg when filled with water to meet the maximum load requirements of the stage (especially at ID19). To minimize the adverse effects of air-bubble formation in terms of imaging and acoustics, the tank was filled with distilled water through a supply valve (inner diameter: 14.5 mm) using a thin rubber hose. The standing wave tube-like tank consisted of a horizontal Plexiglas® tube with an inner diameter of 11 cm, a length of 18 cm (volume: 1.71 l) from shaker-coupled membrane to membrane, and a wall thickness of 5 mm (for further dimensions see Fig 2C). The tank was equipped with a projector (inertial shaker) at each end of the tube (Fig 2A). The shaker-coupled membrane (diameter: 58 mm) at each end of the tube was made of a carbon disk (thickness: 1 mm; diameter: 47 mm) enclosed by a silicone membrane (thickness: 1 mm). The miniature inertial shaker 2002E (PCB Synotech) was coupled to the carbon disk by a steel screw (M3) with a length of 5 cm. During the experiments, the inertial shakers were driven

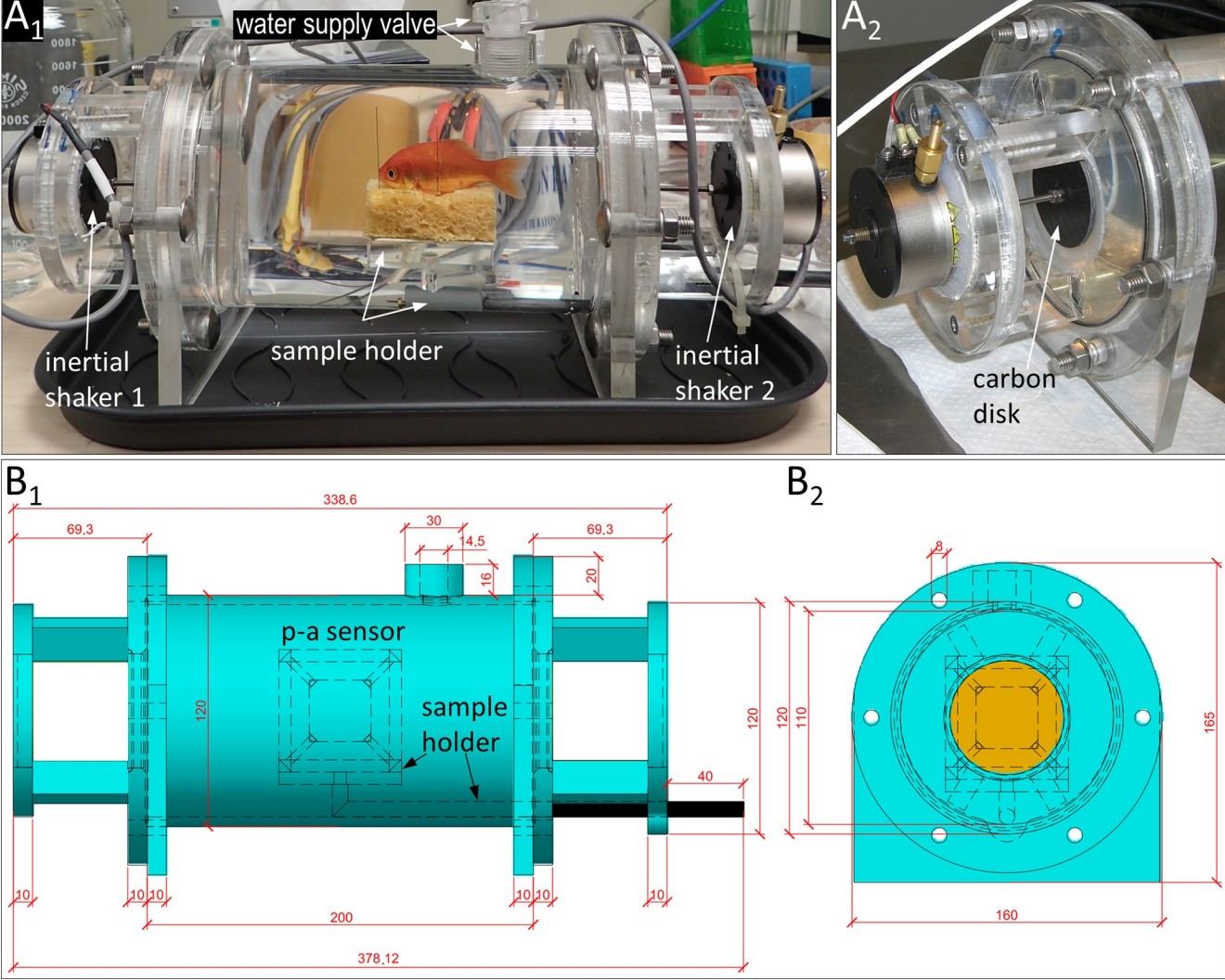

**Fig 2. Overview of the Plexiglas® standing wave tube-like setup. (A₁)** Tank with goldfish in lateral view; the fish is fixed to the foam-covered sample holder using three insect pins. **(A₂)** Detail of the miniature inertial shaker, the carbon disk, and the surrounding silicone membrane. **(B₁-B₂)** Schemes depicting the tank dimensions (inertial shakers not shown). Values in millimetres. B₁-B₂ drawn by Heinz Pfeiffer (University of Vienna).

either in phase (0˚) to create a pressure maximum in the centre of the tank or out of phase (180˚) to obtain maximum particle motion (Fig 2B).

### Sound stimulus and characterization of tank acoustics

**Pure tone stimulus.** Fish were subjected to a 200 Hz pure tone generated by the two inertial shakers. We chose a 200 Hz stimulus because this frequency resulted in a successful observation of the *in situ* motion of otoliths in our last study [30] applying hard X-ray phase contrast imaging at the ESRF. Moreover, this enabled us to compare outcomes of the new setup design with the setup–an upright tank equipped with an underwater speaker–used in our previous study. The stimulus was generated by the software CoolEdit 2000 (Syntrillium Software Corp. Phoenix, AZ) and the internal sound card (Conexant 20561 SmartAudio HD) of a laptop (Lenovo ThinkPad T500) connected to the two inertial shakers via an amplifier (S. M.S.L. SA 36A Pro, Shenzhen ShuangMuSanLin Electronic Co., Ltd., Shenzhen, China). The stimulus was composed of two silent periods (3 s duration in the beginning, 1 s at the end) enclosing the actual sound stimulus (5 s). The 200 Hz stimulus was presented at up to 19 different SPLs ("steps") ranging from 141 up to >177 dB re 1 μPa, four of which were used for the final analysis (Table 1). A certain "step", e.g. step 2, is defined as the same setting of the amplifier and the internal loudspeaker of the laptop during stimulus presentation, regardless whether the shakers were driven in phase or out of phase. Thus, step 2 (at ID19) equals a SPL of 177.2 dB re 1 μPa when shakers were driven in phase and a SPL of 155.4 dB re 1 μPa when shakers were driven out of phase. SPLs decrease from steps 1 to 4 (see Table 1).

**Sound pressure levels.** All measurements to characterize tank acoustics were performed before or after the actual experiments. Simultaneous acoustic measurements were not possible because either the p-a sensor or the miniature hydrophone would have been affected by the X-ray beam, or the SPLs and particle acceleration levels (PALs) would have been evaluated in places other than the tank centre. At both beamlines, SPLs of the different "sound steps" (SPLs, LLSP, L frequency weighting, S time weighting) and of ambient noise present in the beamline hutches (LLeq, L frequency weighting, recording duration: 1 min) were measured using a miniature hydrophone (Brüel & Kjær 8103, sensitivity: -211 dB re 1 V/μPa) connected to a sound level meter (Brüel & Kjær 2238 Mediator) which was calibrated using a hydrophone calibrator (Brüel & Kjær 4229). The ambient noise was recorded with the miniature hydrophone connected to the laptop using the software CoolEdit 2000.

**Particle acceleration levels.** The sound field in the tank was characterized by measuring the SPL directly and calculating the PAL in a walk-in soundproof room at the Department of Behavioral and Cognitive Biology, University of Vienna. The SPL (LLSP) was measured with the miniature hydrophone connected to the sound level meter (details see above). In order to determine PAL in the horizontal direction, a calibrated underwater miniature acoustic pressure-acceleration (p-a) sensor (S/N 2007–001, Applied Physical Sciences Corp., Groton, CT) was placed in the centre of the tank (Fig 2B$_1$). The p-a sensor consists of two built-in units: a piezoelectric, omnidirectional hydrophone (sensitivity: -173.7 dB re 1 V/μPa) and a bi-directional accelerometer (sensitivity: -137.6 dB re 1 V/ μm/s$^2$). The p-output and, subsequently, the a-output of the sensor, was connected to the sound level meter. SPLs were calculated in dB re 1 μPa and PALs in dB re 1 μm/s$^2$. These are the international units for sound pressure and particle acceleration according to ISO standards (ISO 1683, 1983).

**Spectra of the ambient noise and the stimulus.** The ambient noise in the tank was recorded using the software CoolEdit 2000 with a miniature hydrophone (Brüel & Kjær 8103) connected to the internal sound card of the laptop via an amplifier (36B2, constructed at the University of Vienna; including an electrical grounding). The ambient noise spectrum was

then analysed using the sound analysing software STX 3.7.8 (Acoustics Research Institute, Austrian Academy of Sciences, Vienna, Austria).

Tone stimuli were recorded at the Bioacoustics lab (University of Vienna) in a walk-in sound-proof room at three different "sound steps" (step no. 2–4; Table 1). All recordings were performed twice, one when the inertial shakers were driven in phase (0°) and a second time when the shakers were driven out of phase (180°). The stimulus was presented using the same devices, software, and settings described for the experiments performed at ID17 and ID19. To simultaneously record the stimulus, we used the software CoolEdit 2000 and the internal sound card of a second laptop connected to the 36B2-amplifier (again including an electrical grounding).

## Image processing and analysis

Images were processed using ImageJ v. 1.52i (https://imagej.nih.gov/ij/). The 32-bit or 16-bit (unsigned) mode of the original images was retained and the background was removed from each image in the stack by subtraction (stacks from ID19) or division (stacks from ID17) of the averaged reference image (minimum intensity, ID19; average intensity, ID17), which was based on a series of 100–200 raw reference images. The reference images had been recorded without test subject or sound presentation using the same pixel size, frame rate, and integration time. After background correction, brightness and contrast were adjusted and, if necessary, all images of the stack were flipped vertically and/or horizontally to account for virtual flips of the test subject during imaging. To improve the result of the subsequent template matching procedure (see below), image stacks were reduced to 8-bit tif files and subjected to a 3D median filter with a radius of 2 pixels.

The ImageJ plugin "Template matching–Align slices in stack" corrects shifts between images within a stack using a selected region of interest (ROI) as template for slice alignment. The corrections performed in x- and y-directions for each image can be saved as x-y-coordinates. We ran this procedure with the normalized correlation coefficient method in which a match between the template (selected ROI) and a corresponding structure on the subsequent images depends on the relative intensity contrast of the pixels under the template. We chose a uniform size for selected ROIs (40 × 40 pixels; analysis of the motion of the claustrum and the sagitta in goldfish: 20 × 20 pixels), the subpixel registration activated, and a search area around the ROI set to 10–15 pixels. We studied up to 18 ROIs per specimen and orientation (i.e. dorsal, lateral, frontal view). This displacement does not necessarily reflect the true maximum displacement in micrometres of the moving structure within the ROI because those structures within ROIs highly contrasted against the background may show a greater displacement than those that display similar grey values compared to the background. Nonetheless, displacements in micrometres inferred from the pixel size can serve as an estimate of the true displacement. The mean maximum displacement at selected ROIs (i.e. morphological structures that displayed a good contrast against the background) was inferred from the image number versus displacement plots (see S1 File), focusing on the first five oscillations during each sound period. To convert these values into estimates of maximum displacements in micrometres, mean values from these plots were multiplied by the actual pixel size of 3.67 μm (ID19). To facilitate the comparison between different ROIs in the illustrations, linear trends were removed, if present, in the software PAST v. 3.15 using the "Transform–Remove trend" method and, subsequently, residuals were plotted.

In addition to analysing the motion of auditory structures using "Template matching", we produced overlays of the mean minimum and maximum positions of the swim bladder or the Weberian ossicles (tripus) during the motion period using Adobe Photoshop® CS6. To create

an overlay image, ten images of maximum and minimum displacement of the structure of interest were averaged. Then, the averaged minimum and maximum images were copied into the red channel and the green and blue channels of a RGB image, respectively.

Videos (S1–S6 Movies) of representative image stacks from ID17 and ID19 were generated in ImageJ v. 1.52i using a frame rate of 99 fps (ID17) or 198 fps (ID19) with jpeg as compression format. Subsequently, movies in avi format were labelled in Microsoft Powerpoint® and saved as mp4 files. The videos show only a short sequence of ca. 50–100 images for each silent period before and after sound presentation.

### Tomography and 3D reconstruction

In order to characterize structures of interest (otoliths, swim bladder, Weberian ossicles) in a 3D vision, one specimen of each species analysed by 2D radiography experiments was subsequently imaged at ID17 using micro-computed tomography. For this purpose, the fish was put into a 40 ml plastic tube with its snout pointing toward the bottom of the tube. The tube was filled with tap water after the fish had been stabilized with pieces of foam. The imaging was performed at 70 keV, generating 2,600 angular projections equally spaced in a range between 0˚ and 180˚ using an integration time of 50 ms per frame and obtaining a voxel size of 6.1 μm. The goldfish (standard length: 61 mm, body weight: 8.1 g) was scanned, creating three partially overlapping scans (with a maximum height of a single image stack of 4.88 mm) to cover the full range of the otophysic connection and the inner ears. In *E. canarensis* (standard length: 51 mm, body weight: 5.6 g), inner ears and the swim bladder extensions could be captured in a single image stack.

To create one image stack for the goldfish, the three stacks were first merged in Amira® version 6.2 applying "Register images", "Resample transformed images", and "Merge". Subsequently, the otoliths, the Weberian ossicles (goldfish), the anterior parts of the swim bladder as well as parts of the neurocranium, ribs, pharyngeal teeth, and vertebrae were three-dimensionally reconstructed. Structures in the goldfish and the individual of *E. canarensis* were labelled using the threshold tool; a different threshold setting was used for each morphological structure to avoid artificial labelling of adjacent regions with similar grey values that were not part of the structure. If necessary, sprawling structures were erased manually. The labelled structures were resampled. Then, surface rendering was performed by triangulation and the number of faces was reduced to values between $10^5$ and $10^6$ depending on the size and complexity of the structure. To estimate otolith volumes based on surf files, however, surfaces were generated without previous resampling or reducing the number of faces.

## Results

### Performance of the standing wave tube-like setup

In the hutches at the beamlines ID17 and ID19, the ambient noise (background) levels (ANLs) were distinctly lower than the SPLs measured during sound presentation (Table 1). At the same sound step, the in phase condition (0˚) and the out of phase condition (180˚) displayed a difference in SPLs of 14.0–21.8 dB (ID17, ID19) and 24.8–26.0 dB (lab Vienna). The p-a sensor measurements showed that the in phase condition resulted in higher SPLs and lower PALs compared to the values yielded by the out of phase condition (shown for steps 3 and 4). The ambient noise spectrum displayed highest relative amplitudes in the low-frequency range up to ca. 100 Hz (Fig 3A). The sound spectra recorded for both the in phase and the out of phase conditions showed a clear peak at the stimulus frequency, i.e. at 200 Hz (Fig 3B and 3C).

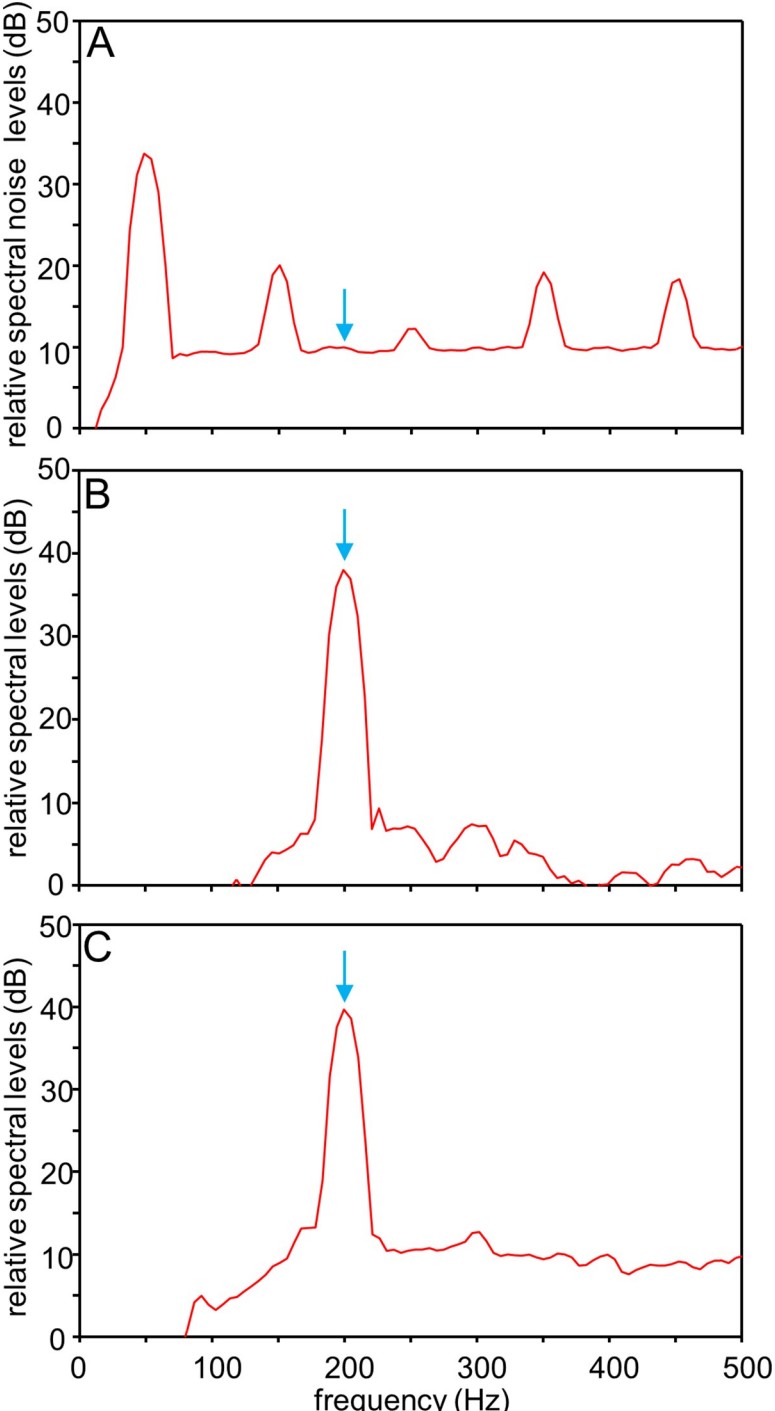

**Fig 3. Noise and tone stimulus spectra recorded in the standing wave tube-like tank at ID19. (A)** Spectrum of background noise. (**B-C**) Spectra of the 200 Hz stimulus recorded in the tank when both inertial shakers were driven in phase (B: 0˚, 177.2 dB re 1 µPa) or out of phase (C: 180˚; 155.4 dB re 1 µPa). The SPL of the 200 Hz stimulus (164.3 dB re 1 µPa (0˚), and 146.2 dB re 1 µPa (180˚)) was distinctly above that measured for the background noise (118.1 dB re 1 µPa). Spectra were analysed at a sampling rate of 44.1 kHz, using a Hanning filter bandwidth of 10 Hz with 75% overlap. Blue arrow indicates stimulus frequency.

## Motion of auditory structures depends on sound pressure and particle motion

The motion patterns of auditory structures in both species clearly differ when the shakers were driven in phase or out of phase. When shakers are driven in phase (pressure maximum in the centre of the tank), the swim bladder walls in *Etroplus canarensis* and goldfish oscillate, whereas no such motion is recorded when shakers are driven out of phase (180˚; S1 and S2 Movies). A similar effect occurs for the tripus, intercalarium and scaphium and the sagitta in goldfish and for the lapillus and sagitta in *E. canarensis*; these structures show distinct motion when subjected to the in phase condition. Under the out of phase condition, structures (especially bones and otoliths) move mainly along the x-axis, which is the axis of the shaker-induced particle motion. Under the in phase condition, structures, especially the saccular and utricular otoliths in *E. canarensis*, move in x-direction but also reveal distinct motion along the y-axis, which corresponds to the dorsoventral (in lateral and frontal view) or mediolateral axis (in dorsal view) of the fish.

## Motion of auditory structures depends on sound levels

A decrease in sound levels (steps 1 to 4) is related to a decrease in maximum displacements of the auditory structures (Fig 4). This decrease in displacement is more distinct when shakers are driven in phase versus out of phase (Fig 4A *versus* 4B). The maximum displacement of structures (e.g. lapillus in *Etroplus canarensis*) along a certain body axis also depends on the lateral, dorsal, or frontal views and the position of the "landmark" (region of interest during the template matching procedure; S1 File).

**Motion in terms of preferred axes of ancillary auditory structures and otoliths.** Here, we describe for each species the motion patterns of the ancillary auditory structures (swim bladder, Weberian ossicles) and the otoliths (especially sagitta and lapilli) under both experimental conditions, i.e. shakers driven in phase (0˚) versus out of phase (180˚). We then briefly characterize the interplay between the moving ancillary auditory structures and the otoliths during the condition with maximum sound pressure at the centre of the tank (in phase condition, 0˚).

**Etroplus canarensis.** The morphology of the anterior swim bladder extensions and the otoliths in *E. canarensis* is very similar to that in *E. maculatus* [26,33]; the dorsally located gas-filled part of the swim bladder closely adjoins the posterior semicircular canal and the recessus located posterior to the utricle (Figs 5A, 5B$_1$, 6A and 6B$_1$).

*Swim bladder motion*. When shakers are driven in phase, the anterior swim bladder extensions show a "complex" motion pattern. Some portions of the swim bladder walls move along the dorsoventral axis while other parts, at the same time, oscillate mainly along the rostrocaudal or mediolateral axes (S2 and S3 Movies). Different parts may also move along the same axis but in the opposite directions (Fig 5B$_2$ and 5B$_3$). The swim bladder walls attached to the vertebral column and parts in the rostroventral portion show weak or no discernible movement (S2 and S3 Movies). When shakers are driven out of phase (180˚), the walls show no clear oscillation but the whole swim bladder moves back and forth mainly along the direction of sound-induced particle motion (Figs 5B$_4$, 5B$_5$–6B$_4$ and 6B$_5$ *versus* 5B$_2$, 5B$_3$, 6B$_2$ and 6B$_3$; S1B Fig *versus* S1A and S1D Fig *versus* S1C).

*Otolith motion*. When shakers are driven in phase, the lapilli move distinctly along all three axes, i.e. the rostrocaudal, the mediolateral, and the dorsoventral axes (Figs 5B$_2$, 5B$_3$–7B$_2$ and 7B$_3$; S1C$_2$ Fig). The sagitta shows motion along the dorsoventral axis mainly in the rostral part of the otolith (S2C$_2$ Fig; S3 Movie) and to a lesser degree along the rostrocaudal axis (S2A$_1$ and S2B$_2$ Fig). The dorsal view reveals that the sagitta tilts medially; this motion is most

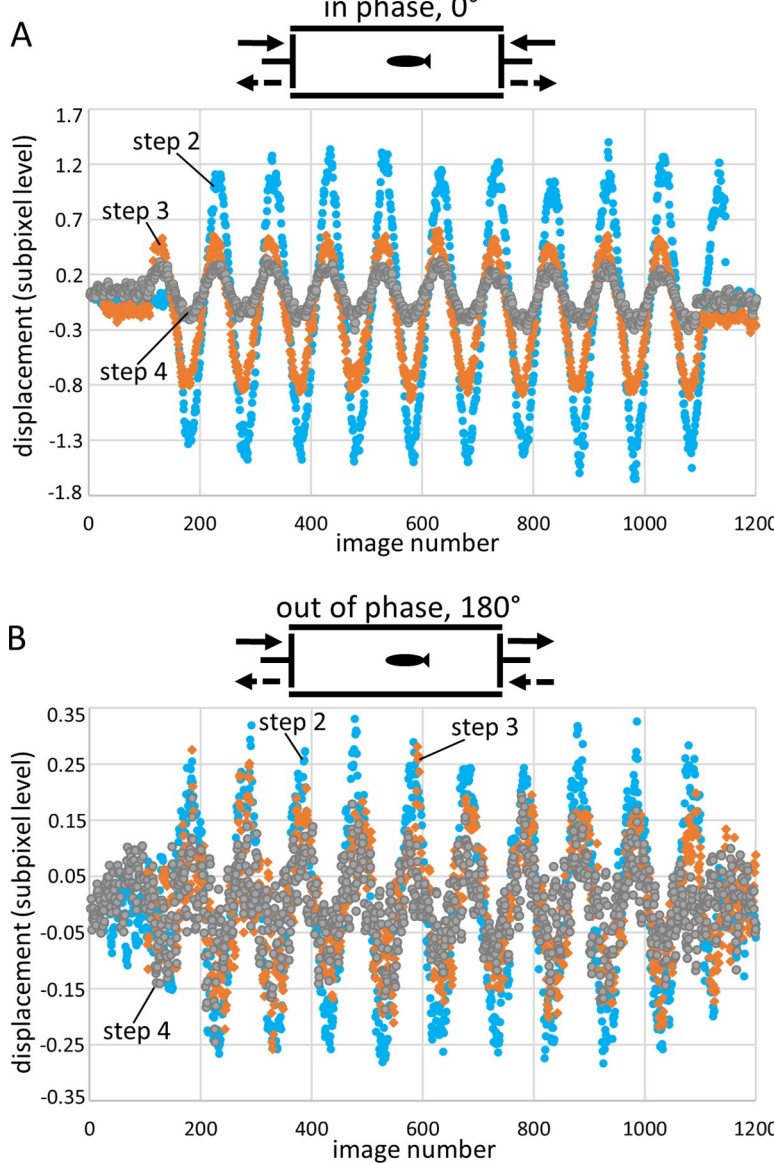

**Fig 4. SPL decrease when shakers are driven in phase versus out of phase.** Maximum displacement along the dorsoventral axis of the utricular otolith (*E. canarensis*) decreases with decreasing SPLs when shakers are driven (**A**) in phase (0°) and (**B**) out of phase (180°). This decrease in maximum displacement is more distinct in (A) than in (B). The scheme above each plot indicates the experimental condition, i.e. the two shakers driven in phase (0°), with pressure maximum at the centre of the tank (A) or shakers driven out of phase (180°), with maximum particle motion at the centre of the tank (B). In (B), the curves of maximum displacement showed a linear trend which was removed in the software Past 3.15 for better illustration. Sound step 2, filled blue circles; sound step 3, orange diamonds; sound step 4, filled grey circles (for corresponding sound pressure and particle acceleration levels see Table 2). Schemes adapted and modified from Hawkins, 1993 [14].

pronounced in the rostral otolith portion, indicated by the phase shift between the anterior tip, the ventral, dorsal, and posterior portions of the sagitta (S2A₂ and S2C₁ Fig). In frontal view, the anterior tip of the sagitta moves in phase with the adjacent part of the neurocranium (S2C₂ Fig). No distinct motion pattern is evident for the asteriscus, which moves slightly along the rostrocaudal and mediolateral axes (Fig 5B₂ and 5B₃). When shakers are driven out of phase (180°), the lapilli and sagitta move distinctly less than during the in phase condition (Figs 5, 6,

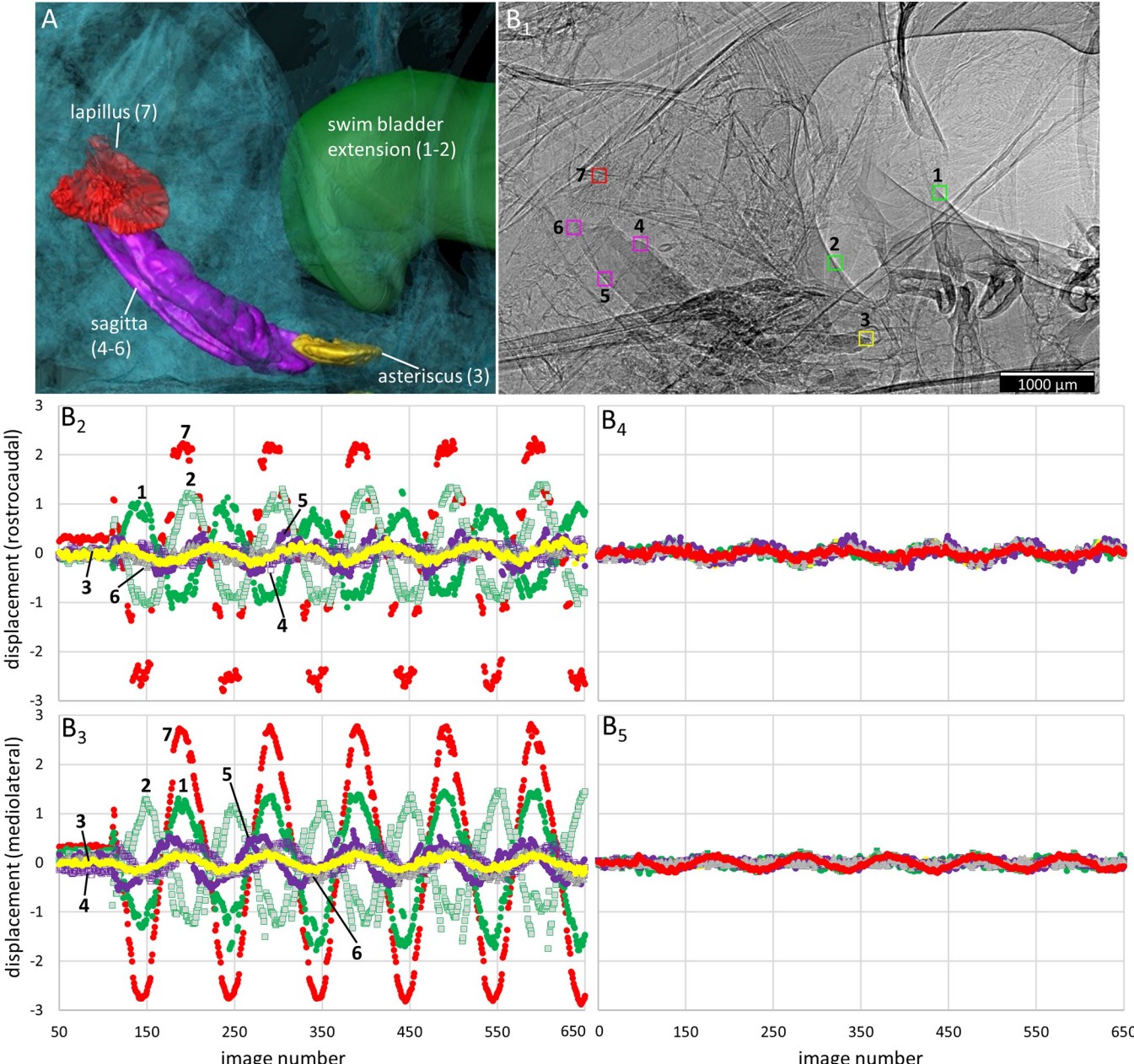

**Fig 5. Motion patterns of otoliths and the anterior swim bladder extensions in *Etroplus canarensis* in dorsal view.** The fish (SL = 59 mm) was subjected to the in phase (0˚, B$_2$-B$_3$) and the out of phase (180˚, B$_4$-B$_5$) conditions (pixel size 3.67 μm, ID19). (**A**) 3D reconstruction of the structures shown in the 2D radiograph in (B$_1$). (**B$_1$**) "Landmarks" (squares of 40 × 40 pixels) that were used to depict the motion of the structures in x- (rostrocaudal; B$_2$, B$_4$) and y- (mediolateral; B$_3$, B$_5$) direction during sound presentation. In both directions (**B$_2$-B$_3$**), the lapillus and the anterior swim bladder extensions moved more than the sagitta and the asteriscus. All structures show distinctly less displacement in the out of phase (**B$_4$-B$_5$**) versus in phase mode (B$_2$-B$_3$).

7B$_4$ and 7B$_5$ *versus* 7B$_2$ and 7B$_3$). Moreover, left and right sagittae, but especially both lapilli, move in the same direction along the mediolateral axis, i.e. left and right otoliths of the same type (e.g. both lapilli) move either to the left or the right side (S3 Movie).

*Interplay of swim bladder and otolith motion (0°, in phase condition).* When the ventral wall parts of the swim bladder extensions move in ventral direction and the rostral-most part of the

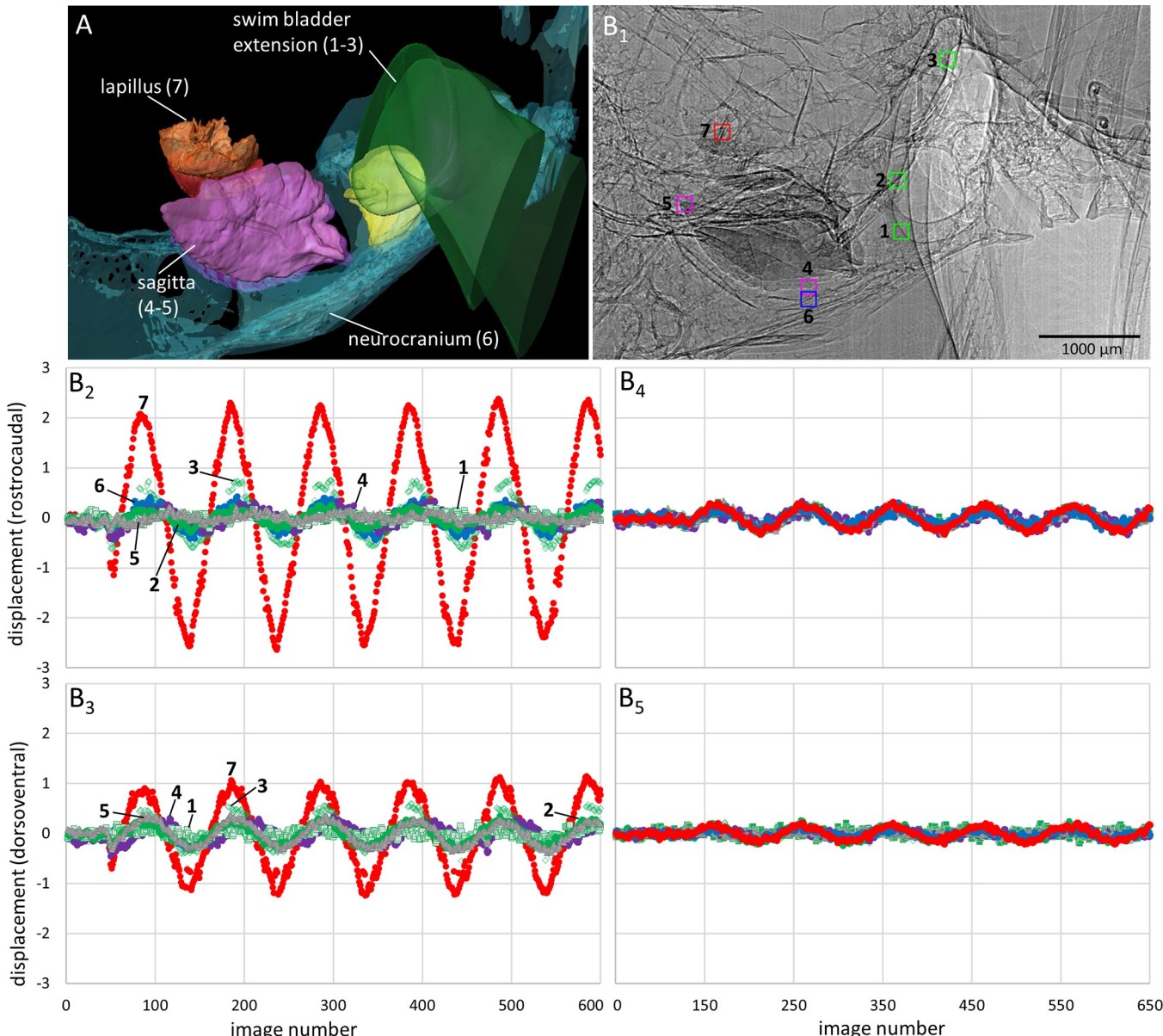

**Fig 6. Motion patterns of otoliths and the anterior swim bladder extensions in *Etroplus canarensis* in lateral view.** The fish (SL = 48 mm) was subjected to the in phase (0°, $B_2$-$B_3$) and the out of phase (180°, $B_4$-$B_5$) conditions (pixel size 3.67 μm, ID19). (**A**) 3D reconstruction of the structures shown in the 2D radiograph in ($B_1$). ($B_1$) "Landmarks" (squares of 40 × 40 pixels) that were used to depict the motion of the structures in x- (rostrocaudal) and y- (dorsoventral) direction during sound presentation. ($B_2$-$B_3$) In both directions, the lapillus moved more than the sagitta and asteriscus or the swim bladder extensions. All structures show distinctly less displacement in the out of phase ($B_4$-$B_5$) versus in phase mode ($B_2$-$B_3$).

extensions move rostrally, the lapilli move at the same time (dorso)rostrally and laterally while the sagittae move ventrally and tilt medially.

**Goldfish.** *Motion of the swim bladder and Weberian ossicles.* When the shakers are driven in phase (0°), the anterior swim bladder walls of the goldfish oscillate along the rostrocaudal and dorsoventral axes (Fig $8C_1$ and $8D_1$; $S3C_1$ and $S3D_1$ Fig). The tripus and the intercalarium move mainly along the rostrocaudal and dorsoventral axes (Fig $8D_1$; $S3C_1$ and $S3D_1$ Fig), whereas the scaphium also displays a distinct motion along the mediolateral axis ($S3C_2$ and $S3D_2$ Fig; S4 Movie), i.e. towards the sinus impar. In lateral view, the scaphium moves more

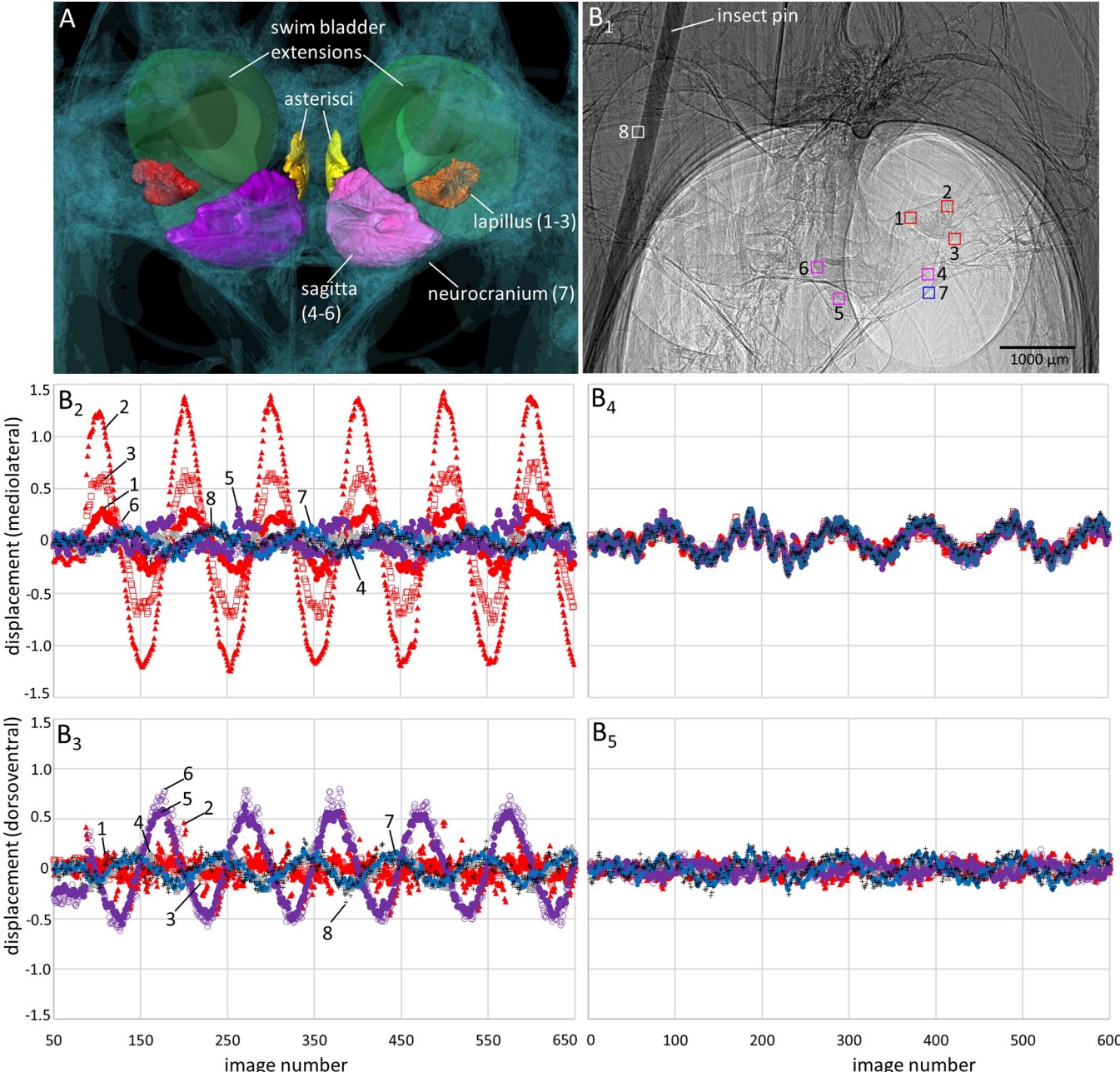

**Fig 7. Motion patterns of otoliths and the anterior swim bladder extensions in *Etroplus canarensis* in frontal view.** The fish (SL = 49 mm) was subjected to the in phase (0˚, $B_2$-$B_3$) and the out of phase (180˚, $B_4$-$B_5$) conditions (pixel size 3.67 μm, ID19). (**A**) 3D reconstruction of the structures shown in the 2D radiograph in ($B_1$). (**$B_1$**) "Landmarks" (squares of 40 × 40 pixels) that were used to depict the motion of the structures in x- (mediolateral) and y- (dorsoventral) direction during sound presentation. In (**$B_2$**), the lapillus shows distinctly more motion than the sagitta and asteriscus or the swim bladder extensions along the mediolateral axis, whereas in ($B_3$) the sagitta moves most along the dorsoventral axis. All structures are displaced distinctly less in the out of phase (**$B_4$-$B_5$**) versus in phase condition (**$B_2$,$B_3$**).

pronouncedly along the dorsoventral axis than the claustrum ($S4C_1$ and $S4C_2$ Fig, $S5B_3$ and $S5C_3$ Fig; S5 Movie). In the out of phase condition (180˚), the swim bladder walls show no clear oscillation but the whole swim bladder moves mainly along the axis of sound-induced particle motion (Fig $8C_2$ and $8D_2$; $S3C_3$,$S3C_4$ and $S3D_3$ Fig, but see $S3D_4$ Fig). The Weberian

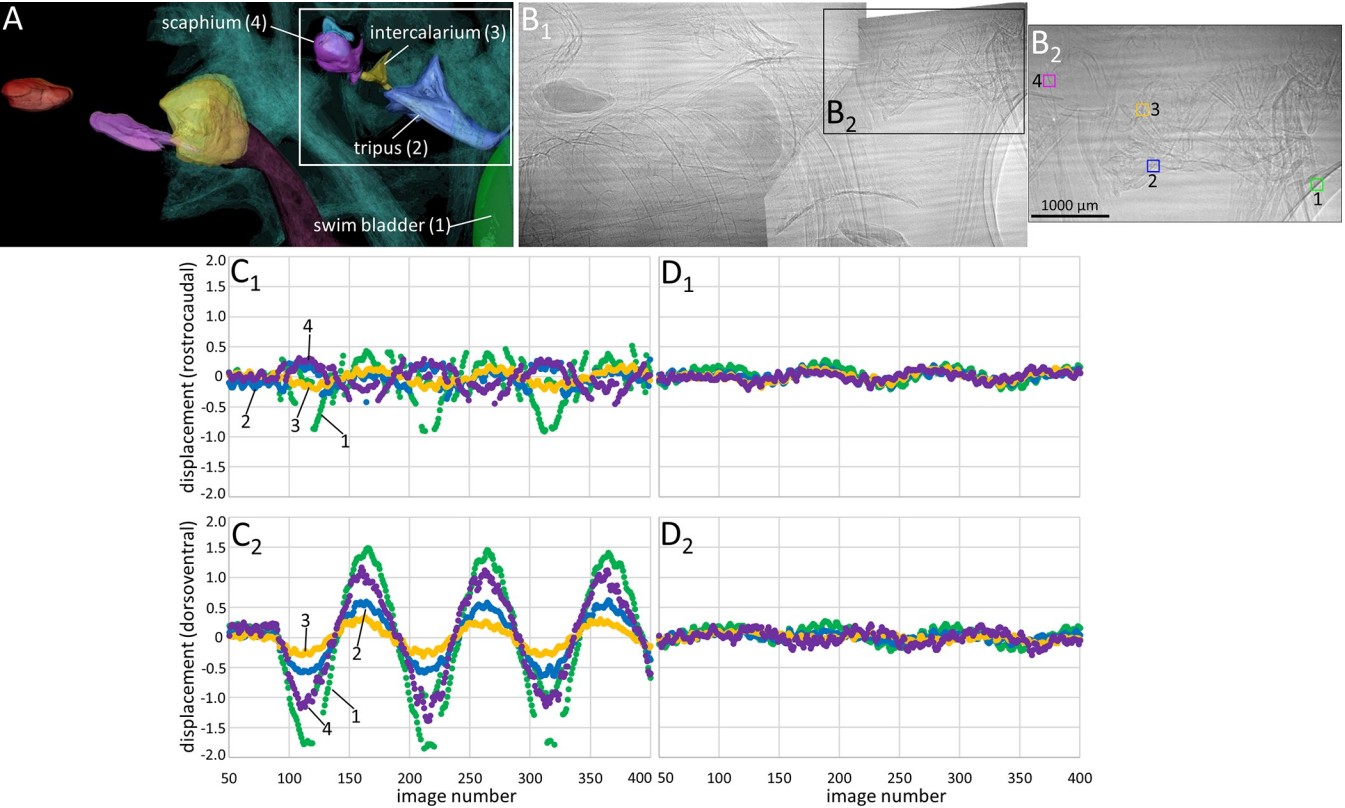

**Fig 8. Motion patterns of the Weberian ossicles (scaphium, intercalarium, tripus) and the anterior-most swim bladder portion in goldfish in lateral view.** The fish (SL = 50 mm) was subjected to the in phase (0˚, $C_1$-$C_2$) and the out of phase (180˚, $D_1$-$D_2$) conditions (pixel size 3.67 μm, ID19). (**A**) 3D reconstruction of the structures shown in the 2D radiograph in ($B_1$). ($B_{1-2}$) "Landmarks" (squares of 40 × 40 pixels) depict the motion of the structures in x- (rostrocaudal) and y- (dorsoventral) direction during sound presentation. The Weberian ossicles and the anterior swim bladder wall show distinctly less displacement when the inertial shakers are driven out of phase ($D_1$-$D_2$) versus in phase ($C_1$-$C_2$).

ossicles also move distinctly less than in the in phase condition (Fig $8C_2$ and $8D_2$ *versus* $8C_1$ and $8D_1$; $S3C_3$, $S3C_4$, $S3D_3$ and $S3D_4$ Fig *versus* $S3C_1$, $S3C_2$, $S3D_1$ and $S3D_2$ Fig, S4B Fig *versus* S4A Fig).

Comparing the left and right Weberian ossicles (S3 Fig) reveals that the amount of displacement, and whether or not a phase shift occurs, differs between left and right ossicles of the same type (see also S1 File for the scaphium).

*Otolith motion.* When shakers are driven in phase (0˚), the sagitta moves along the rostrocaudal axis ($S6C_1$ Fig, $S7C_1$, $S7C_3$ and $S7D_1$ Fig), tilts laterally ($S7C_2$, $S7C_4$ and $S7D_2$ Fig; S6 Movie) and shows distinct motion along the dorsoventral axis (Fig $9C_2$ *versus* $9C_1$). The lateral tilt of the sagitta was also clearly visible when the fish was subjected to a 500 Hz pure tone stimulus (frame rate = 497.12 fps, SPL = 177.3 dB re 1μPa; $S7D_1$ and $S7D_2$ Fig). The asterisci and lapilli display (weak) motion along all three axes and the amount of motion is similar for both conditions (0˚ and 180˚; Fig 9; S6 Fig); however, lapilli seen in dorsal view show a stronger rostrolateral movement under the out of phase versus the in phase condition. The sagitta does not display a clear lateral tilt under the out of phase condition and its displacement along the dorsoventral axis is distinctly smaller than under the in phase condition (Fig $9C_2$ *versus* $9D_2$).

*Interplay of the motion of all auditory structures (0°, in phase condition).* When the rostral swim bladder walls move in (dorso-)rostral direction, the scaphium moves rostrodorsally and simultaneously in medial direction, while the sagitta shifts also rostrodorsally and at the same

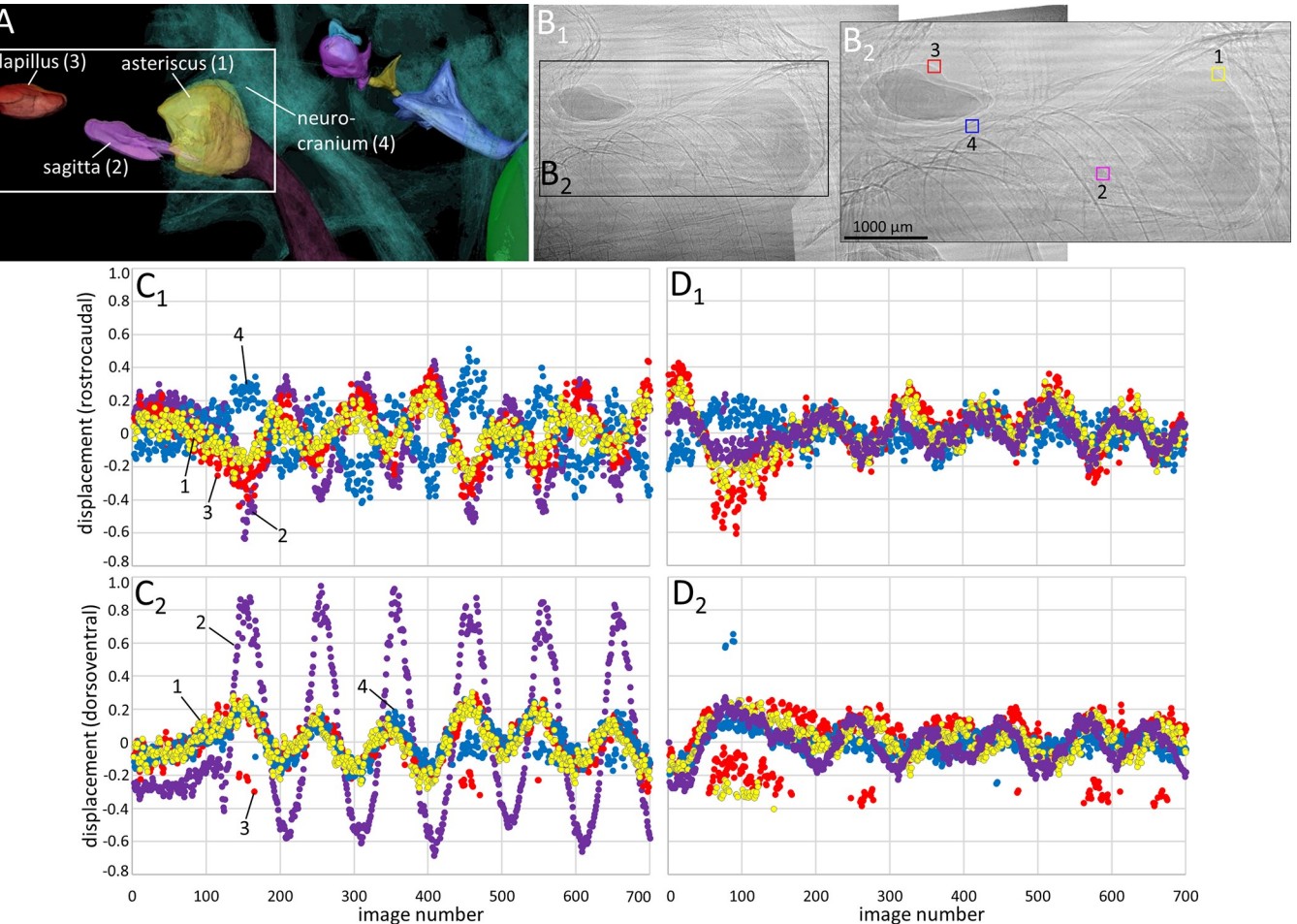

**Fig 9. Motion patterns of the otoliths (lapillus, sagitta, asteriscus) in goldfish in lateral view.** The fish (SL = 50 mm) was subjected to the in phase (0˚, **C₁-C₂**) and the out of phase (180˚, **D₁-D₂**) conditions (pixel size 3.67 µm, ID19). (**A**) 3D reconstruction of the structures shown in the 2D radiograph in (B₁). (**B₁₋₂**) "Landmarks" (squares of 40 × 40 pixels) depicted the motion of the structures in x- (rostrocaudal) and y- (dorsoventral) direction during sound presentation. The maximum displacement of lapilli and asterisci is similar regardless of whether the inertial shakers are driven in phase or out of phase. Along the dorsoventral axis, the sagitta shows less displacement in the out of phase condition (C₂ *vs.* D₂).

**Table 2. Otolith volume and estimated otolith mass in the two specimens subjected to tomography at ID17.**

| *Species* | *C. auratus* (SL = 61 mm) | | *E. canarensis* (SL = 51 mm) | |
|---|---|---|---|---|
| *Otolith type* | V [mm³] | m [mg] | V [mm³] | m [mg] |
| left asteriscus | 1.397 | 3.55 | 0.079 | 0.20 |
| right asteriscus | 1.445 | 3.67 | 0.077 | 0.20 |
| left sagitta | 0.082 | 0.24 | 0.632 | 1.85 |
| right sagitta | 0.100 | 0.29 | 0.635 | 1.86 |
| left lapillus | 0.751 | 2.20 | 0.147 | 0.43 |
| right lapillus | 0.732 | 2.15 | 0.112 | 0.33 |

Volume data derived from 3D reconstructions of otoliths generated in the software Amira® 6.2. Estimation of otolith mass based on the density of biogenic aragonite (2.93 g/cm³: sagittae, lapilli; [49]) and biogenic vaterite (2.54 g/cm³: asterisci; [50]).

time tilts laterally. The movie (S5 Movie) shows that the sagitta tilts laterally when the scaphium moves medially towards the sinus impar.

**Relationship between otolith size and otolith motion.** Comparing otolith motion in both species reveals weak motion for large otoliths (asterisci, lapilli) in goldfish and light otoliths (asterisci) in *E. canarensis* (Table 2). In *E. canarensis*, lapilli move more (1.2–8.7 μm at SPL = 164.3 dB re 1 μPa) than sagittae (0.6–2.2 μm at SPL = 164.3 dB re 1 μPa). In goldfish, the lapilli are displaced less (1.0–1.8 μm at SPL 164.3 dB re 1 μPa) than the sagittae (0.6–3.9 μm at SPL 164.3 dB re 1μPa; S1 File).

When shakers are driven out of phase (180˚), the difference in displacement between otolith types is much smaller; the amount of motion of sagittae and lapilli of *E. canarensis* ranges from 0.7–1.2 μm at SPL 146.2 dB re 1μPa and for that in goldfish ranges from 0.5–1.0 μm (sagittae) and 1.2–1.5 μm at SPL 146.2 dB re 1μPa (lapilli; S1 File).

## Discussion

We successfully improved our initial setup presented in Schulz-Mirbach et al., 2018 [30] to separately study the motion of auditory structures based on sound pressure and sound-induced particle motion. First, we present models of how otolith motion patterns may stimulate different groups of ciliary bundles on the respective macula and compare the potential effects of two different types of otophysic connections on otolith motion. Our discussion focusses on the utricular and saccular otoliths because the lagenar otoliths in both species showed amplitudes too small to be reliably interpreted. In the second part, we discuss methodological aspects of the new setup.

### Assumed interplay between the moving auditory structures and inner ear stimulation

**Etroplus canarensis.** The oscillating walls of the dorsal, gas-filled part of the swim bladder extension in *E. canarensis* may enhance the endolymph motion in the recessus posterior to the utricle (see Fig 9C in [26]), thereby enhancing lapillus motion. This is in line with the concerted forward movement of the lapillus and the swim bladder wall close to the recessus, and with a previously hypothesized stimulation of the utricle (see Fig 9C in [26]). The "swinging" motion of the lapillus may stimulate most of the sensory hair cells on the cotillus and the striola of the macula utriculi (Fig 10D; see Fig 4C in [26]). The stronger motion of the lapillus due to the contribution of the swim bladder (in phase condition) may also stimulate the sensory hair cells on the lacinia (= dorsolateral macula portion). Although the lapillus does not overlie the lacinia, a (strongly) back-and-forth "swinging" otolith might exert enough drag force on the otolithic membrane connecting otolith and macula to each other (including the lacinia), thereby resulting in a deflection of the ciliary bundles of the lacinia (see pers. comm. A. N. Popper in [26]).

The sagitta motion pattern may be explained by the attachment of the anterior tip of the saccule to the surrounding bone [26], as the tip has distinctly fewer degrees of freedom to move along the dorsoventral axis than the "free" posterior part of the saccule. Accordingly, the sagitta shows a greater dorsoventral displacement in its posterior region (Fig 10A, 10C and 10D). As most ciliary bundles on the posterior portion of the macula sacculi are vertically oriented (Fig 10D, see also Fig 5C in [33]), the strong sagitta motion in this region should stimulate the sensory hair cells. Both the anterior and posterior-most portion of the macula sacculi are characterized by horizontally oriented ciliary bundles and the slight movement of the sagitta along the rostrocaudal axis (Fig 10C) might stimulate the sensory hair cells here. The tilting motion might be provoked by endolymph passing through the thin canal (see Fig 3C in

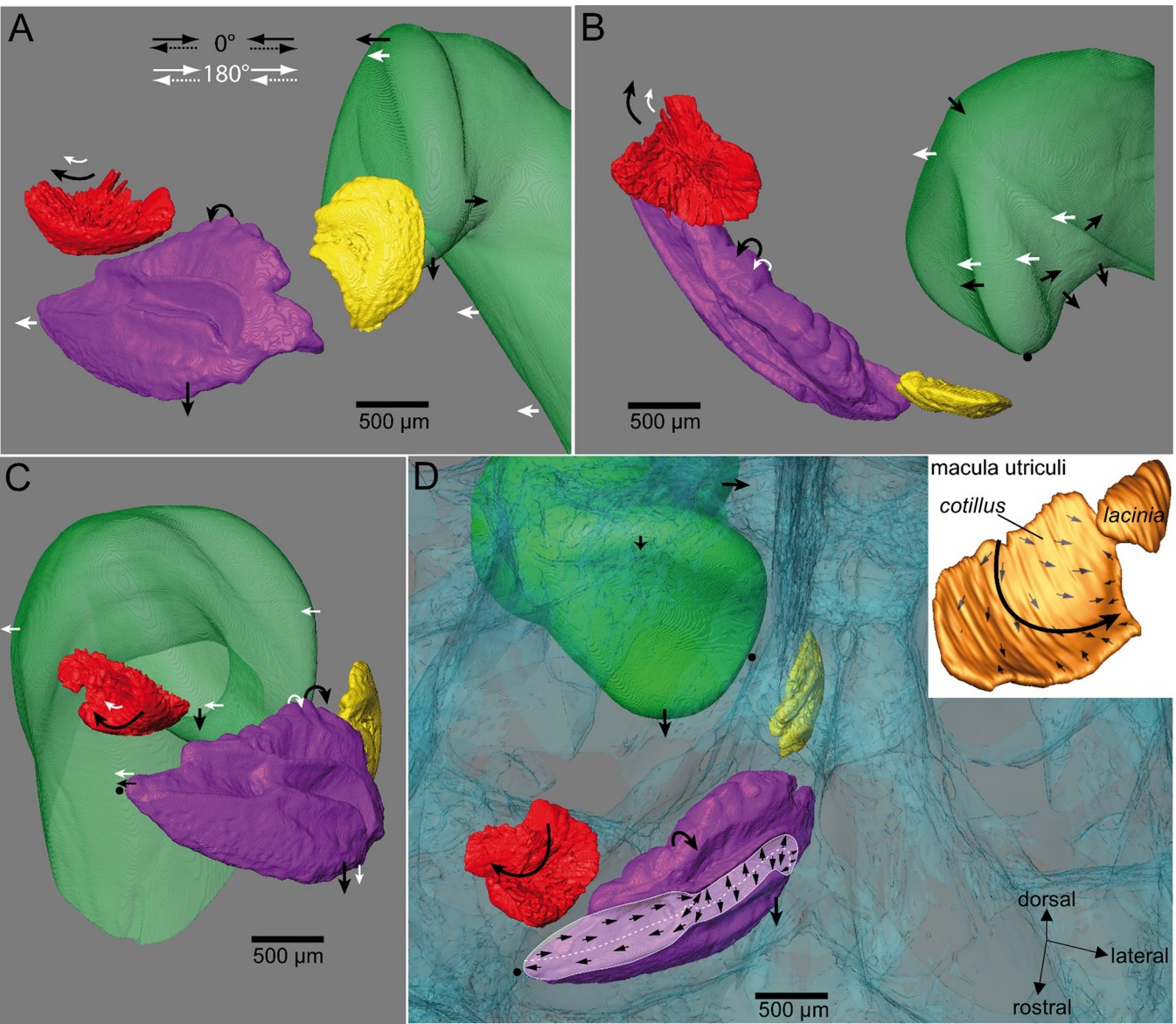

**Fig 10. Model of otolith and swim bladder motion in *Etroplus canarensis*.** Auditory structures are shown in (**A**) medial, (**B**) dorsal, (**C**) frontal, and (**D**) a perspective view. In (D), structures are shown in rostral, medial, and dorsal aspects to visualize the assumed 3D motion of the lapillus, sagitta, and the anterior swim bladder extension (0˚ condition) also with regard to the underlying maculae. Black arrows: motion when fish was subjected to the in phase condition (0˚); white arrows: out of phase condition (180˚). For clarity, 3D reconstructions of only the otoliths and the swim bladder extension of the right side are shown in (A), (B), and (C). Green: anterior swim bladder extension; red: lapillus; purple: sagitta; yellow: asteriscus, blue: bone. Shape of maculae and the orientation patterns of ciliary bundles adapted and modified from Schulz-Mirbach et al., 2014 [33].

[26]) connecting the saccule to the upper ear and thereby pushing the dorsal-most hump of the sagitta against the macula sacculi. The lateromedial tilt of the sagitta (Fig 10A–10D) may stimulate 1) the sensory hair cells in the "neck" of the macula sacculi, in which ciliary bundles are diagonally oriented with respect to the rostrocaudal axis and 2) those of the rostrodorsal portion covered by otolithic membrane only (see Fig 3C in [26]). In the latter case, the lateral tilt along with a backwards motion (along the rostrocaudal axis) may pull the otolithic membrane backwards and diagonally to the rostrocaudal axis resulting in a deflection of ciliary bundles not overlain by the otolith. This is in line with theoretical considerations on the

stimulation of sensory hair cells not overlain by the otolith (for an overview see [4]). Whether the oscillation of the swim bladder walls is also transmitted to the neurocranium (and further transferred to the saccule at the contact site between saccular tip and the bone) remains elusive. Our data do not provide support for this pathway of local particle motion transduction (see [26]): the displacement of the neurocranium surrounding the saccule was similar for both conditions, i.e. regardless of whether the fish was subjected to the in phase (0˚) or the out of phase (180˚) condition.

When fish were studied in lateral or dorsal view, the main axis of the sound corresponded to the fish's rostrocaudal axis and left and right sagittae and lapilli showed the same synchronous motion. Accordingly, if fish were alive, the relative motion between otolith and the underlying macula would be the same for the left and the right side regardless of the experimental condition applied (0˚ or 180˚). The situation is more complex when sound impinges on the body flanks of the fish ("experiments in frontal view"). For the in phase condition (0˚), left and right sagittae and lapilli showed the same (mirror-imaged) motion. Again, this means that sensory hair cells on the maculae are stimulated in the same way on the left and the right side. For the out of phase condition (180˚), however, left and right lapilli both moved simultaneously either to the left or to the right side. In this case, the respective orientation groups of ciliary bundles on the left and right maculae utriculi do not experience the same shearing force. Hence, a combination of otolith motion provoked by sound-induced particle motion and sound pressure would result in a different stimulation of left and right otolith end organs (here: utricle and saccule). This interpretation is in accordance with models of the direct and indirect stimulation pathway in fish with an otophysic connection (see Fig 5 in [16]). The potential role of differences in motion patterns between left and right otoliths for the ability of fishes to localize a sound source is, however, beyond our data and remains to be studied.

**Goldfish.** The motion of the tripus, intercalarium, and scaphium provoked by the oscillating anterior swim bladder walls agrees with the movement of these ossicles hypothesized by previous studies [44,51–53]. We can also confirm the assumption [54] that the claustrum "does not" move and thus may act as a counter-bearing to the scaphium, which displays a distinct motion. In this way, the perilymphatic fluid in the part of the sinus impar enclosed by the scaphium and the claustrum would efficiently be pressed into the main anterior portion of the sinus impar. This then results in a corresponding fluid flow in the sinus endolymphaticus and the transverse canal (Fig 11B and 11C).

Earlier studies [44,52,53] predicted that fluid flow of the endolymph through the dorso (medially) located transverse canal into each of the saccules should "impinge" on the free (lateral) wing of the sagitta. If this was the case, the sagitta should display a lateral tilt, which is the motion we observe in our experiments with goldfish subjected to the in phase (0˚) condition (Fig 11B). The fluid flow from the dorsal margin to the lateral wing of the sagitta may also enhance the dorsoventral motion of this otolith. The distinct dorsoventral motion of the sagitta probably stimulates the sensory hair cells, which possess mainly vertically oriented ciliary bundles (Fig 11A; [55]). The tilting motion may also lead to the mediolateral motion of the posterior needle-like part of the sagitta. In this posterior region of the sagitta, the macula sacculi is curved, bringing the initially vertically oriented ciliary bundles into a more mediolateral orientation [55]. Hence, sagitta motion in the posterior region would stimulate the mediolaterally oriented ciliary bundles.

In contrast to the sagitta, which shows a greater displacement under the in phase (0˚) condition, the motion of the lapillus is more distinct in the out of phase (180˚) condition. This supports the hypothesis of a more pronounced division of labour between the otolith end organs in otophysans [45]. This hypothesis states that the otophysan saccule is an end organ specialized to detect sound pressure, whereas the otoliths of the lagena and the utricle act as

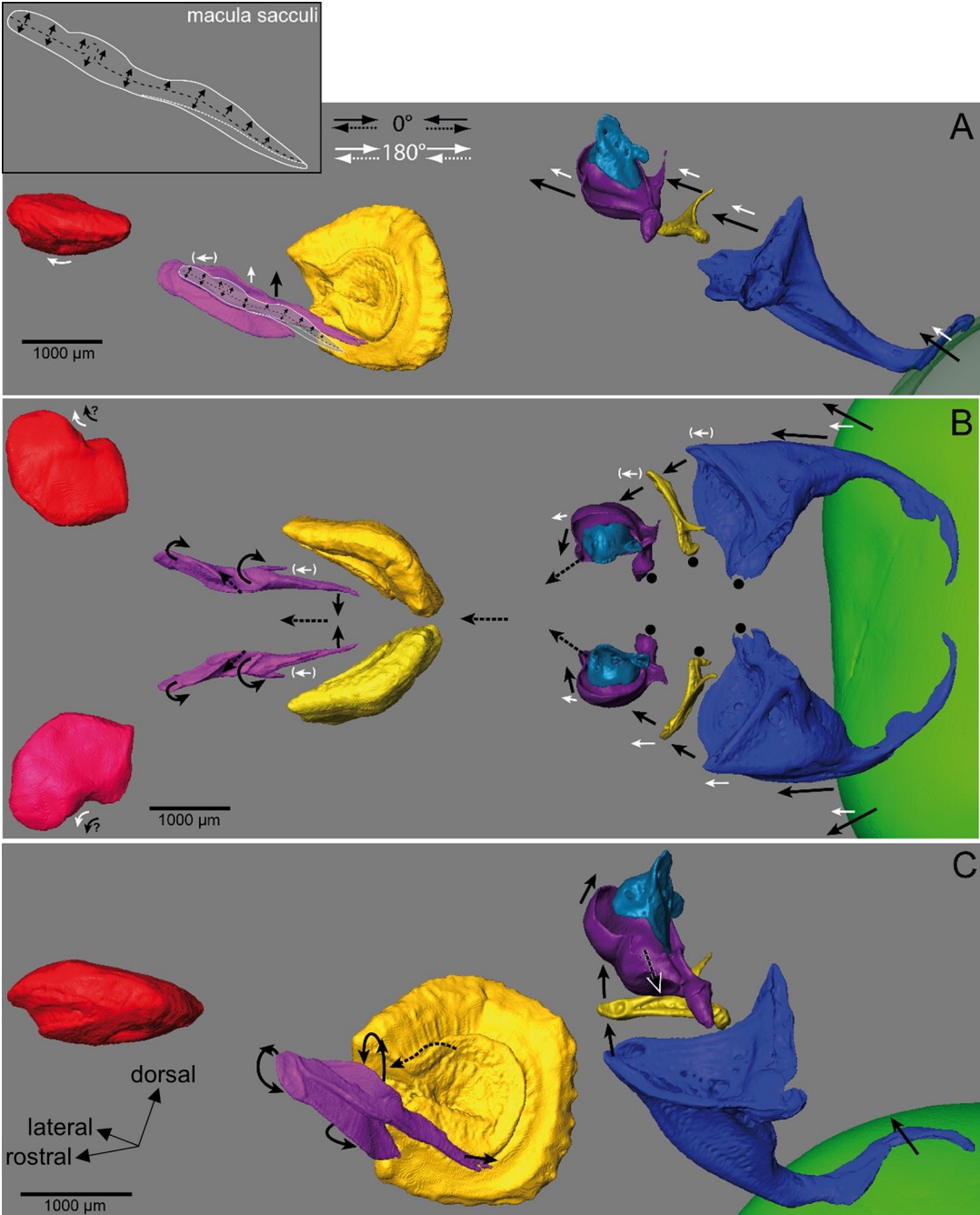

**Fig 11. Model of otolith, Weberian ossicle, and swim bladder motion in goldfish.** Auditory structures are shown in (A) lateral, (B) dorsal, and (C) a perspectival view. In (**A**), a 3D scheme of the macula sacculi including the orientation pattern of ciliary bundles is shown in context with the overlying sagitta. In (**B**) and (**C**), black arrows with dotted lines indicate the motion of the hypothesized perilymphatic fluid in the sinus impar and the endolymphatic fluid in the transverse canal and the saccules (see [44,51–53]). In (C), structures are shown in rostral, medial, and dorsal aspects to visualize the assumed 3D motion of the sagitta, the Weberian ossicles, and the swim bladder (0° condition). Black arrows: motion under the in phase condition (0°), white arrows the out of phase condition (180°). White arrows in parentheses: no relative motion of structure with regard to the surrounding bones (neurocranium, vertebrae). For clarity, 3D reconstructions of the otoliths and the Weberian ossicles of only the right side are shown in (A) and (C). Green: swim bladder (anterior-most part); red: lapillus; purple: sagitta; yellow: asteriscus, light blue: claustrum; dark purple: scaphium; gold: intercalarium; dark blue: tripus. Shape of macula sacculi and orientation pattern of ciliary bundles in (A) adapted and modified from Platt, 1977 [55].

accelerometers detecting sound-induced particle motion. In our experiments, the lapillus motion was generally very small, making it difficult to determine whether there is a distinct difference in lapillus motion between the two conditions applied, i.e. 0˚ *versus* 180˚.

## Otolith mass and the type of otophysic connection apparently affect otolith motion

Comparing the estimated mass of the three otolith types in goldfish and *E. canarensis* (Table 2) with the observed motion patterns of these otoliths clearly reveals that the amount of otolith motion does not follow the simple correlation of the greater the mass the greater the displacement as predicted by mathematical modelling (see for a 200 Hz stimulus Fig 10 in [56]). In the two studied species that represent two different types of otophysic connections, otolith motion clearly depends on other additional factors. The otophysic connection enhances otolith motion in both species; nonetheless, different otolith types are affected to a different amount within the same species and also show species-specific differences. The motion of the lapillus and the sagitta in *E. canarensis* is clearly boosted when the swim bladder walls oscillate under the in phase (0˚) condition, whereas in goldfish, only sagitta motion is enhanced by swim bladder wall oscillation and Weberian ossicle motion. Moreover, the light sagitta in goldfish moves distinctly, whereas the similarly light asteriscus in *E. canarensis* shows only faint movement. Finally, the "heavy" sagitta in *E. canarensis* moves pronouncedly, whereas the heavier lapillus and asteriscus in goldfish reveal no or only weak motion.

One potential reason for the species-specific differences is the amount of damping effect of the otophysic connection. The system in *E. canarensis* may enable more overmodulation at the high SPLs applied in our experiments because the anterior swim bladder extensions directly contact the thin bony lamella close to the lagena and near the exoccipital foramina approaching the posterior part of the upper ears [26]. For the otophysan type of otophysic connection (goldfish), the system is more damped because the saccules are indirectly connected to the swim bladder through the chain of Weberian ossicles [29]. Accordingly, in a more damped system, the local particle motion transmitted to accelerometer-like otoliths such as the lapillus and the asteriscus is less than in the *Etroplus*-type; the motion of lapilli and asterisci may be so small that it is at or beyond the limits of the spatial resolutions available in our experiments. Finally, our findings provide further support for the assumption that the otophysan saccule acts as a displacement sensor [29] in which sagitta motion mainly depends on the incoming fluid flow from the transverse canal. Sagitta motion seems to increase with increasing sound frequency based on mathematical modelling of the function of the Weberian apparatus in goldfish [29]. It remains to be studied whether the sagitta in *E. canarensis* is similarly influenced by fluid flow through the thin canal connecting the upper ear with the saccule.

In *E. canarensis*, the light asteriscus is likely to act as an accelerometer, and its mass may be too low to show a distinct motion due to the 200 Hz pure tone stimulus. At frequencies above 200 Hz, when the swim bladder contribution increases, greater local particle motion may be transmitted to the asteriscus, resulting in a stronger motion of this otolith. Note that a recent study on the plainfin midshipman (*Porichthys notatus* Girard, 1854) [57], a species that lacks an otophysic connection, indicates that auditory thresholds of the lagena (based on sound pressure and particle acceleration) were higher than that of the saccule in the frequency range from 85 to 505 Hz.

In both species, sagitta motion was enhanced through the oscillation of the swim bladder walls when fish were subjected to the in phase (0˚) condition. Such an enhancement of otolith motion at low frequencies in goldfish agrees with previous experimental studies on tripus extirpation and swim bladder deflation [39,58] as well as with mathematical modelling of the

motion of the auditory structures [29]. A swim bladder effect at 200 Hz was confirmed by Ladich & Wysocki, 2003 [58] for goldfish based on bilateral tripus extirpation. A mathematical model further predicts that the Weberian pathway of saccule stimulation has an enhancement effect over the entire range of detectable frequencies (i.e. above 100 Hz; [29]). However, in *Etroplus maculatus*, AEP thresholds did not differ significantly in the range from 100 to 300 Hz [15] indicating a minor swim bladder effect at 200 Hz, similar to that measured in the brown bullhead *Ameiurus nebulosus* (Lesueur, 1819) (Otophysa, Ictaluridae) during swim bladder deflation experiments [59]. Swim bladder deflation experiments in *Etroplus maculatus* and *E. canarensis* may help clarify the degree to which the swim bladder contributes to audition at 200 Hz.

## Potential and limitations of the standing wave tube-like setup

Compared to the one-speaker setup developed by Schulz-Mirbach et al. (2018) [30], relative movements between different otolith types (sagitta and lapillus in *E. canarensis*), bones, and the swim bladder walls were clearly visible using the current standing wave tube-like setup (displacements in this study: up to 8.7 μm at 164 dB re 1 μPa; in [30]: ca. 6.5–13.0 μm at 158 dB re 1 μPa). The difference of at least 20 dB for measured SPLs between the in phase (0˚) and out of phase condition (180˚) points to a largely successful separation of the particle and pressure components of sound in our setup. This was also clearly reflected in the difference in motion of the auditory structures between these two conditions. In addition, we were able to characterize the motion patterns because of the higher spatial resolution of 3.67 μm available at ID19 versus 6.5 μm in the former study [30].

Nonetheless, several issues remain. The compliance of the Plexiglas® walls with a thickness of only 5 mm may be a problem regarding the in phase condition. Achieving a real pressure maximum with "no" particle motion at the centre of the tank would require using more rigid and stiff walls that do not oscillate during sound presentation. This can only be realized with a steel tank at a wall thickness of ca. 1 cm (see [17]). In our setup, wall thickness could not be increased because this would also substantially increase the attenuation of the X-ray beam, resulting in a poor signal-to-noise ratio [30]. Another problem of tank acoustics is related to the samples themselves. Hawkins & MacLennan (1976) [17] pointed out that the gas-filled swim bladder of a fish may alter the sound field, and to avoid this issue the authors studied a flatfish species lacking internal gas bladders (plaice, *Pleuronectes platessa* L., 1758). In our experiments this aspect might be, however, negligible because both studied species possess a large, two-chambered swim bladder [15].

From an imaging point of view, the "large" water body in the tank impairs a real-time observation of the moving auditory structures in the fish. At ID19, maximum image acquisition rates of up to 1 kHz (1,000 fps) can be achieved, which would be sufficient to record these motions in real time. However, such high frame rates are only possible for samples investigated in air [60,61]. It is thus important to note that our observations of moving otoliths and ancillary auditory structures are a proxy of the true motion.

High sound pressure levels and tissue damage by hard X-ray imaging might be another source of artefacts. The sound pressure levels used in our study are distinctly higher than those experienced by fish in natural habitats [42,62,63]. Yet, we are not able to detect otolith movement at a few nanometers (e.g. at auditory thresholds) due to the constraints implied by the imaging procedure. However, as the motion of the otoliths displayed a regular and reproducible pattern for different SPLs (see S1 File), we assume that the observed patterns may also hold true for biologically relevant SPLs.

To minimise detrimental effects of e.g. changed elasticity of the tissue, we avoided to conduct experiments in the same order. A comparison of histological sections of fish subjected to

hard X-rays with those of control fish in future studies could help to evaluate the kind and degree of tissue damage of the inner ears and the swim bladder.

## Conclusions

Even though further methodological improvements such as higher temporal resolutions are envisaged, the current setup is a suitable compromise between the intended separation of sound-induced particle motion and sound pressure on the one hand and the requirements of the X-ray based imaging procedure on the other. Our setup enabled the detailed characterization of the interplay between the otoliths and ancillary auditory structures such as anterior swim bladder extensions and Weberian ossicles. This yielded extensive experimental evidence for hypotheses on how different types of otophysic connections affect otolith motion. We were also able to draw first conclusions on how the motion patterns of the sagitta and the lapillus (the latter only in *E. canarensis*) stimulate the sensory hair cells on the respective macula.

The 200 Hz stimulus (0˚) provoked a motion pattern of the sagitta in both species and the lapillus in *Etroplus canarensis* that closely fit the orientation patterns of ciliary bundles on the macula sacculi or the macula utriculi. In *E. canarensis*, we conclude that the specific otolith motion patterns stimulate most of the orientation groups on the respective macula under both experimental conditions (0˚ and 180˚). The stimulation of sensory hair cells on both the macula sacculi and the macula utriculi is probably enhanced in *E. canarensis* because the swim bladder acts as pressure-to-displacement transducer due to the resulting greater displacement of the sagitta and the lapillus. In goldfish, the lapillus was displaced more when subjected to the out of phase (sound-induced particle motion) condition, whereas the sagitta moved more under the in phase (sound pressure) condition. We conclude that, in goldfish, this difference in otolith motion under the in phase and the out of phase condition is experimental support for a hypothesized division of labour of otolith end organs in otophysans, in which the saccule apparently plays a highly specialized role in sound pressure detection [45].

## Supporting information

**S1 File. Amount of displacement of otoliths and ancillary auditory structures.** Representative output data from the template matching analysis performed in ImageJ v. 1.52i of both fish species are given, including displacement plots. Data depict motion patterns of selected "landmarks" (regions of interest 40 × 40 pixels) related to sound steps 2 to 4 for both conditions, i.e. 0˚ and 180˚. Displacements in micrometres are given based on a pixel size of 3.67 μm (ID19). (XLSX)

**S1 Movie. Motion of the utricular and saccular otoliths and the swim bladder extensions in** *Etroplus canarensis* **in lateral view.** The fish was subjected to a 200 Hz pure tone stimulus when inertial shakers were driven in phase (0˚). (frame rate: 99 fps; ID17, pixel size 6.1 μm). (MP4)

**S2 Movie. Motion of the utricular and saccular otoliths and the swim bladder extensions in** *Etroplus canarensis* **in lateral view.** The fish was subjected to a 200 Hz pure tone stimulus when inertial shakers were driven in phase (0˚) or out of phase (180˚), respectively. (frame rate: 198 fps; ID19, pixel size 3.67 μm). (MP4)

**S3 Movie. Motion of the utricular and saccular otoliths and the swim bladder extensions in** *Etroplus canarensis* **shown in frontal view.** The fish was subjected to a 200 Hz pure tone stimulus when inertial shakers were driven in phase (0˚) or out of phase (180˚), respectively.

(frame rate: 198 fps; ID19, pixel size 3.67 μm).
(MP4)

**S4 Movie. Motion of the Weberian ossicles and the swim bladder in goldfish shown in dorsal view.** The fish was subjected to a 200 Hz pure tone stimulus when inertial shakers were driven in phase (0˚). (frame rate: 198 fps; ID19, pixel size 3.67 μm).
(MP4)

**S5 Movie. Motion of left and right scaphia and claustra in goldfish shown in lateral view.** The fish was subjected to a 200 Hz pure tone stimulus when inertial shakers were driven in phase (0˚). (frame rate: 198 fps; ID19, pixel size 3.67 μm).
(MP4)

**S6 Movie. Motion of the otoliths (mainly the sagittae) in goldfish shown in dorsal view.** The fish was subjected to a 200 Hz pure tone stimulus when inertial shakers were driven in phase (0˚). (frame rate: 198 fps; ID19, pixel size 3.67 μm).
(MP4)

**S1 Fig. Motion patterns of auditory structures in *Etroplus canarensis* in lateral view.** Overlays of averaged maximum (red) and minimum (green) positions indicate the motion of otoliths (lapilli, sagittae) and the walls of the anterior swim bladder extensions due to the stimulus presentation. (**A**) and (**C**) depict the structures during the in phase (0˚) condition of the sound stimulus, whereas (**B**) and (**D**) represent the out of phase (180˚) condition. ($C_2$) *vs.* ($D_2$) illustrates that the swim bladder walls show distinct motion (shakers driven in phase) as opposed to no oscillation (shakers driven out of phase). Structures outlined in red and green indicate motion, whereas uniformly gray, white or black structures point to no or weak movement during sound presentation. The overlays in (A-B) represent an individual (SL = 51 mm) studied at ID17 (pixel size 6.1 μm, frame rate 98.9954 fps) and those in (C-D) illustrate a specimen (SL = 48 mm) imaged at ID19 (pixel size 3.67 μm, frame rate 198.02 fps).
(TIF)

**S2 Fig. Motion patterns of the rostral tip as well as the ventral, dorsal, and caudal margins of the sagitta in *Etroplus canarensis*.** Motion patterns are shown in ($\mathbf{A_{1\text{-}2}}$) dorsal, ($\mathbf{B_{1\text{-}2}}$) lateral, and ($\mathbf{C_{1\text{-}2}}$) frontal views (pixel size 3.67 μm, ID19) when the fish was subjected to the in phase (0˚) condition. The clear phase shift between different parts of the sagitta in all plots except in ($B_2$) indicates the tilting movement of this otolith during sound presentation. In ($C_1$), the rostral tip of the sagitta shows less displacement than the ventral and posterior margins and moves in phase with the adjacent part of the neurocranium, which is also seen in ($B_1$). The respective landmark number is given in parenthesis (see also Figs $5B_1$, $6B_1$ and $8B_1$).
(TIF)

**S3 Fig. Motion patterns of the Weberian ossicles (scaphium, intercalarium, tripus) and the anterior-most swim bladder portion in goldfish in dorsal view.** The fish (SL = 56 mm) was subjected to the in phase (0˚, $C_1$-$C_2$, $D_1$-$D_2$) and the out of phase (180˚, $C_3$-$C_4$, $D_3$-$D_4$) conditions (pixel size 3.67 μm, ID19). (**A**) 3D reconstruction of the structures shown in the 2D radiograph in ($B_1$). ($\mathbf{B_{1\text{-}2}}$) "Landmarks" (squares of $40 \times 40$ pixels) depict the motion of the structures in x- (rostrocaudal) and y- (mediolateral) direction on the right (**C**) and left (**D**) body side during sound presentation. The Weberian ossicles and the anterior swim bladder wall show distinctly less displacement when the inertial shakers are driven out of phase ($C_3$-$C_4$, $D_3$-$D_4$) versus in phase ($C_1$-$C_2$, $D_1$-$D_2$).
(TIF)

**S4 Fig. Motion patterns of auditory structures in goldfish in (pseudo-)lateral view.** Overlays of averaged maximum (red) and minimum (green) positions indicate the motion of otoliths (sagittae), Weberian ossicles, and the walls of the anterior swim bladder portion due to stimulus presentation. (**A**) and (**C**) depict the structures during the in phase (0˚) condition of the sound stimulus, (**B**) during the out of phase (180˚) condition. ($C_2$) illustrates that tripus, intercalarium, and scaphium move distinctly while the motion of the claustrum is weak. Structures outlined in red and green indicate motion, whereas uniformly grey, white, or black structures point to no or weak movement during sound presentation. (A-B) represent an individual (SL = 59 mm) studied at ID17 (pixel size 6.1 μm, frame rate 98.9954 fps) and C illustrates a specimen (SL = 50 mm) imaged at ID19 (pixel size 3.67 μm, frame rate 198.02 fps).
(TIF)

**S5 Fig. Motion pattern of scaphium and claustrum in two different individuals of goldfish in lateral view.** The fishes (B, SL = 50 mm; C, SL = 58 mm) were subjected to the in phase (0˚) condition. (**A**) 3D reconstruction of the structures shown in the 2D radiograph in ($B_1$) and ($C_1$). In ($\mathbf{B_1, C_1}$), "landmarks" (squares of $20 \times 20$ pixels) depict the motion of the structures in x- (rostrocaudal) and y- (dorsoventral) direction during sound presentation. Along the rostrocaudal axis ($\mathbf{B_2, C_2}$), both ossicles show a similar amount of displacement (except landmark 4 in $C_2$). Along the dorsoventral axis ($B_3, C_3$), the scaphium in both individuals is displaced more than the claustrum.
(TIF)

**S6 Fig. Motion patterns of the otoliths (lapillus, sagitta, asteriscus) in goldfish in dorsal view.** The fish (SL = 56 mm) was subjected to the in phase (0˚, $\mathbf{C_1}$-$\mathbf{C_2}$) and the out of phase (180˚, $\mathbf{D_1}$-$\mathbf{D_2}$) conditions (pixel size 3.67 μm, ID19). (**A**) 3D reconstruction of the structures shown in the 2D radiograph in ($B_1$). ($\mathbf{B_{1-2}}$) "Landmarks" (squares of $40 \times 40$ pixels) depicted the motion of the structures in x- (rostrocaudal) and y- (mediolateral) direction during sound presentation. Otoliths show a greater maximum displacement along the rostrocaudal than along the mediolateral axis. The maximum displacement of lapilli and asterisci is similar regardless of whether the inertial shakers are driven in or out of phase.
(TIF)

**S7 Fig. Motion pattern of the sagitta in goldfish in dorsal view.** The fish (SL = 56 mm) was subjected to the in phase condition (0˚) using a 200 Hz (SPL = 177.2 dB re1 μPa; frame rate = 198.02 fps; C) or a 500 Hz pure tone stimulus (SPL = 177.3 dB re 1 μPa; frame rate = 497.512 fps; D) (pixel size 3.67 μm, ID19). (**A**) 3D reconstruction of the structures shown in the 2D radiograph in (B). (**B**) "Landmarks" (squares of $20 \times 20$ pixels) depict the motion of the structures in x- (rostrocaudal) and y- (mediolateral) direction during sound presentation. In ($\mathbf{C_1}$-$\mathbf{C_2}$), landmarks 1, 4, and 6 show a phase shift with regard to landmarks 2–3 and 5–7, indicating a tilting motion of the sagitta. In ($\mathbf{C_3}$-$\mathbf{C_4}$) and ($\mathbf{D_1}$-$\mathbf{D_2}$) this phase shift is shown for the landmarks 1 and 3 also with respect to the motion of the adjacent bone (landmark 8).
(TIF)

## Acknowledgments

We are grateful to Anthony D. Hawkins (Aberdeen, UK) for advice regarding the tank design, Heinz Pfeiffer (University of Vienna) for tank construction, and Andrzej Szpetkowski (University of Vienna) for technical support. We thank Alexander Rack for technical support at ID19, Alexandra Demory for support in the lab at ID17, and Loïc de Saint Jean for animal care

at ID17. Bernhard Ruthensteiner (Bavarian State Collection of Zoology, Munich) helped with the stack merging procedure in the software Amira®. Michael Stachowitsch is acknowledged for scientific English proofreading. We also like to thank the editor Dennis M. Higgs and two anonymous reviewers for constructive comments on an earlier version of this article.

## Author Contributions

**Conceptualization:** Tanja Schulz-Mirbach, Friedrich Ladich, Martin Heß.

**Data curation:** Tanja Schulz-Mirbach, Martin Heß.

**Formal analysis:** Tanja Schulz-Mirbach, Alberto Mittone, Margie Olbinado, Martin Heß.

**Funding acquisition:** Tanja Schulz-Mirbach, Friedrich Ladich, Petr Krysl, Martin Heß.

**Investigation:** Tanja Schulz-Mirbach, Friedrich Ladich, Alberto Mittone, Margie Olbinado, Alberto Bravin, Isabelle P. Maiditsch, Roland R. Melzer, Martin Heß.

**Methodology:** Tanja Schulz-Mirbach, Friedrich Ladich, Alberto Mittone, Margie Olbinado, Petr Krysl, Martin Heß.

**Project administration:** Tanja Schulz-Mirbach, Martin Heß.

**Visualization:** Tanja Schulz-Mirbach, Martin Heß.

**Writing – original draft:** Tanja Schulz-Mirbach, Friedrich Ladich, Martin Heß.

**Writing – review & editing:** Tanja Schulz-Mirbach, Friedrich Ladich, Martin Heß.

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
