## [Decision Letter · Decision Letter 0]

13 Jan 2020

PONE-D-19-30473

Auditory chain reaction: effects of sound pressure and particle motion on auditory structures in fishes

PLOS ONE

Dear Dr. Schulz-Mirbach,

Thank you for submitting your manuscript to PLOS ONE. After careful consideration, we feel that it has merit but does not fully meet PLOS ONE’s publication criteria as it currently stands. Therefore, we invite you to submit a revised version of the manuscript that addresses the points raised during the review process.

Please see the reviewers' comments and my detailed summary of these comments below.

We would appreciate receiving your revised manuscript by Feb 27 2020 11:59PM. To enhance the reproducibility of your results, we recommend that if applicable you deposit your laboratory protocols in protocols.io, where a protocol can be assigned its own identifier (DOI) such that it can be cited independently in the future. For instructions see: http://journals.plos.org/plosone/s/submission-guidelines#loc-laboratory-protocols

We look forward to receiving your revised manuscript.

Kind regards,

Dennis M. Higgs

Academic Editor

PLOS ONE

Additional Editor Comments (if provided):

As you can see, both reviewers were highly complemented of the value of this work and had relatively small corrections to be made before publication. Both reviewers commented on the rather high sound levels used in the experiments and I do agree with them that are are of little direct "ecological" relevance as such but do recognize they may have been needed to accurately measure the fine-scale movements in the current paper. A short note in the manuscript stating this and arguing as to whether or not there would be a linear damping of the movements with a decrease in sound levels would be helpful to rebut any criticisms in this regard. The reviewers also note that the discussion is overly long so effort should be made to reduce length here to increase readability. In particular I find you take too much space pointing out all the limitations to your study (MS pages 15 & 16) and this could be greatly reduced to maybe even one paragraph and perhaps moved to later in the discussion section. Other parts of the discussion could also be profitably condensed. Reviewer 2 also notes that the large number of figures further reduced readability so consider whether any of those could also be removed.

Reviewers' comments:

Reviewer's Responses to Questions

**Comments to the Author**

1. Is the manuscript technically sound, and do the data support the conclusions?

Reviewer #1: Yes

Reviewer #2: Yes

2. Has the statistical analysis been performed appropriately and rigorously? 

Reviewer #1: Yes

Reviewer #2: I Don't Know

3. Have the authors made all data underlying the findings in their manuscript fully available?

Reviewer #1: Yes

Reviewer #2: Yes

4. Is the manuscript presented in an intelligible fashion and written in standard English?

Reviewer #1: Yes

Reviewer #2: Yes

5. Review Comments to the Author

Reviewer #1: Review of Schulz-Mirbach et al. manuscript titled “Auditory chain reaction: effects of sound pressure and particle motion on auditory structures in fishes”.

The manuscript is a very well written paper that investigates sound-induced motion of the otoliths within the fish inner ear end organs from two fish that have different otophysic connections of the swim bladder to the inner ear. This paper is a follow up study to their previous work but in the current paper the authors a standing wave tube-like tank to examine the motion of the inner ear otoliths under conditions where there is maximum particle motion vs maximum sound pressure. The authors provide the first experimental evidence of how the two different types of otophysic connections affect otolith motion in the fish inner ear. The subject of the paper should be of general interest to the readers of Plos One. I have only a few comments below.

1) The stimulus used to induce motion of the otoliths within the end organs was quite high ranging from 141 to >177 dB re 1 microPa. The authors reported (in the methods lines 206-207) that “step (at ID19) equals a SPL of 177.2 dB re 1 microPa when shakers were in phase and a SPL of 155.4 re 1 microPa when shakers were driven out of phase”. These sound levels seem to be out of the biologically relevant ranges. I realize that these levels were likely required in order to visualize the induced motion of the otoliths. My concern is that the induce motion and how it may activate motion of the inner ear may not be biologically relevant to natural auditory stimuli experienced by these fish in nature. To alleviate such concerns, the authors should at least address this issue and explain the limitations (I know they have to some degree in the discussion) of how more natural sound levels might induce similar motions at lower levels.

2) An interesting note on line 399-401 that states that the left and right Weberian ossicles revealed differences in the amount of displacement when subjected to sound induced motion. The authors also describe the differential motion of the otoliths based on experimental condition. Are the authors suggesting that the Weberian apparatus might afford directional sensitivity to the sound source? This would be extremely interesting if true. Also, their description of the sound induced movement of the otoliths in the Canara pearlspot cichlid may have implications for how the inner ear uses the particle motion and indirect sound pressure cues to determine sound source direction via the phase model hypothesis.

3) The authors also describe other interesting results of the “tilting motion” of the sagitta in E. canarensis. Was there any evidence for the so-called “rocking motion” of otoliths under their experimental conditions as described by Krysl et al (2012) in PloSOne.

4) Surprisingly the authors do not discuss the induced movement of the asteriscus in E. canarenis given the close proximity of the lagena to the swim bladder in the cichlid. Although sound induce movement was minimal at the experimental sound levels used, movement of this otolith could be very important in terms of the indirect detection of particle motion from the swim bladder. Fay reported that fish can detect particle motion displacement as low as 1 nm. Thus, the authors should not discount the importance the lagena and saccule given their close proximity to the swim bladder.

Reviewer #2: The study using a new apparatus to image the auditory portions of the inner ear during both sound pressure and particle motion. The design is clever and is novel in that it could image this structures during stimulation. The authors do a good job of indicating potential limitations with their procedures. The study makes a very nice overall contribution to the field.

A couple of items that need to be clarified or considered.

The fish are dead and tissue structure can break down very quickly, especially if high intensity lighting was needed and it generated heat. It would have been useful if there was postmortem examination to determine if there were any structure changes that could not be detected by imaging. For example, what was the effect of blood clotting?

The SPL were very high and not normally used in fish auditory studies. Was this due to the limitations in the imaging? I would expect to see large scale movements at these SPL that fish may never experience. How did this impact the studies and models?

I presume the test chamber is designed to get around the problems associated with auditory studies in small tanks. Can physiological studies eventually be done in these types of tanks?

I felt the discussion is far too speculative. The authors have a very nice data set and do not need to explain every variance that may be impacting the structure. I would suggest that the discussion focuses on what can definitely be determined and limit the number of speculative discussion.

I realize this is an on line journal and page limits are not a concern, however 18 figures seem a little excessive and it is hard for the reader to wade through and determine which ones are really need. I would suggest that several figures could be moved to supplemental files.

Minor

Abstract The use of the term maximum even in quotations is confusing.

29 “inner” ears. Many times the authors use the term ear, when inner ear would be more accurate

40 I think it is safe to say they are no longer “assumed”

45-46. Sentences end and start with same words

What is an improved hearing ability. This is usually termed increased sensitivity or greater frequency range

51 … whole body in motion: It is unclear the effect of hard skeletal elements. Also, the motion is oscillolatory with no net movement gain and therefore motion is not best word? Vibration?

78 The pearlspot is not considered as well investigated as the goldfish

95 The saccule has plays a very important vestibular function and therefore does not “mainly” function in audition

113 The cessation of opercular movement is not considered an endpoint of euthanasia. Many fish (including goldfish) can be easily revived after the cessation of opercular movement. There should have been an additional period following the cessation of movement to insure complete euthanasia

Methods

The image rate of 200 Hz for the 200 Hz seems relatively slow. Most video is run higher than stimulus. Please explain if this had an effect on resolving the movements

633 add reference

Table 1: delete, add info to text

6. PLOS authors have the option to publish the peer review history of their article (what does this mean?). If published, this will include your full peer review and any attached files.

Reviewer #1: No

Reviewer #2: No

---

## [Author Response · Author response to Decision Letter 0]

17 Feb 2020

Response to additional comments made by the editor:

High sound pressure levels: We carefully addressed the criticism regarding high SPLs. We think that linear damping of otolith movements with a decrease in SPLs cannot be concluded based on our data because the decrease in (otolith) oscillations was larger at higher SPLs (177.2 down to 170.7 dB) than at lower levels (170.2 down to 164.3 dB; see also please S1 File). Yet, we are not able to detect otolith movement at a few nanometers (e.g. at auditory thresholds) due to the constraints implied by the imaging procedure. However, as the motion of the otoliths displayed a regular and reproducible pattern for different SPLs (e.g. 177.2, 170.7, 164.3 dB re 1 µPa), we assume that the observed patterns may also hold true for biologically relevant SPLs.

Discussion: As recommended, we shortened the discussion by removing or shortening several paragraphs (word count, original ms version: 3,301; revised version: 2,636). In addition, we moved the (shortened) part discussing the methodology to the end of the Discussion chapter. However, to address the reviewers’ comments with regard to potential effects of tissue damage and high SPLs, we included two new (but short) paragraphs.

Figures: We removed 7 figures to the Supporting information resulting in 11 instead of 18 figures in the main text.

Response to reviewers:

Reviewer #1: 

The manuscript is a very well written paper that investigates sound-induced motion of the otoliths within the fish inner ear end organs from two fish that have different otophysic connections of the swim bladder to the inner ear. This paper is a follow up study to their previous work but in the current paper the authors a standing wave tube-like tank to examine the motion of the inner ear otoliths under conditions where there is maximum particle motion vs maximum sound pressure. The authors provide the first experimental evidence of how the two different types of otophysic connections affect otolith motion in the fish inner ear. The subject of the paper should be of general interest to the readers of Plos One. I have only a few comments below.

Thank you very much for the positive feedback on our ms. 

1) The stimulus used to induce motion of the otoliths within the end organs was quite high ranging from 141 to >177 dB re 1 microPa. The authors reported (in the methods lines 206-207) that “step (at ID19) equals a SPL of 177.2 dB re 1 microPa when shakers were in phase and a SPL of 155.4 re 1 microPa when shakers were driven out of phase”. These sound levels seem to be out of the biologically relevant ranges. I realize that these levels were likely required in order to visualize the induced motion of the otoliths. My concern is that the induce motion and how it may activate motion of the inner ear may not be biologically relevant to natural auditory stimuli experienced by these fish in nature. To alleviate such concerns, the authors should at least address this issue and explain the limitations (I know they have to some degree in the discussion) of how more natural sound levels might induce similar motions at lower levels.

We address this point in the Discussion to a larger extent than previously. We completely agree that the high SPLs used in our study are of no biological significance but they are necessary to detect otolith oscillations in situ using hard X-ray imaging. Spatial and temporal resolutions at tomography beamlines are constantly improved. Hence, it may be possible to characterize otolith motion at much lower SPLs and higher frequencies in future studies.

2) An interesting note on line 399-401 that states that the left and right Weberian ossicles revealed differences in the amount of displacement when subjected to sound induced motion. The authors also describe the differential motion of the otoliths based on experimental condition. Are the authors suggesting that the Weberian apparatus might afford directional sensitivity to the sound source? This would be extremely interesting if true. Also, their description of the sound induced movement of the otoliths in the Canara pearlspot cichlid may have implications for how the inner ear uses the particle motion and indirect sound pressure cues to determine sound source direction via the phase model hypothesis.

The different motion patterns of otoliths depending on experimental conditions might have implications for sound source localization. However, we refrained from discussing such a possibility because the topic of explaining the mechanisms of how fish may be able to unambiguously localize a sound source is still vividly debated. As we used high SPLs and as our setup cannot provide a 100% separation of sound-induced particle motion and sound pressure, we felt it being too speculative to interpret our data with regard to sound source localization.

3) The authors also describe other interesting results of the “tilting motion” of the sagitta in E. canarensis. Was there any evidence for the so-called “rocking motion” of otoliths under their experimental conditions as described by Krysl et al (2012) in PloSOne.

Based on our 2D radiographic imaging, it is difficult to decide whether or not otoliths such as the sagitta in E. canarensis may show a “rocking motion”. Under these imaging conditions, we observe a moving auditory structure only from a certain view often overlain by other skeletal elements like cranial bones. Capturing the motion of the otoliths in their full 3D aspect in future experiments using tomography is likely to allow for a more precise characterization of the motion pattern and may clarify whether the movement is a rocking motion sensu Krysl and colleagues.

4) Surprisingly the authors do not discuss the induced movement of the asteriscus in E. canarenis given the close proximity of the lagena to the swim bladder in the cichlid. Although sound induce movement was minimal at the experimental sound levels used, movement of this otolith could be very important in terms of the indirect detection of particle motion from the swim bladder. Fay reported that fish can detect particle motion displacement as low as 1 nm. Thus, the authors should not discount the importance the lagena and saccule given their close proximity to the swim bladder.

We agree that the close proximity of the lagena to the swimbladder extensions in Etroplus may indicate a more prominent role in audition of this otolith end organ. However, due to a rather weak contrast of asterisci compared to sagittae and lapilli (see please Fig. 6B1 vs. A) – we could hardly identify the asterisci in the radiographs when the fish was seen in lateral and frontal views – in combination with a weak overall motion of asterisci, it was very difficult to characterize the motion pattern of the lagenar otoliths. Hence, we think that our data do not allow for discussing or interpreting the motion of the lagenar otoliths.

Reviewer #2: 

The study using a new apparatus to image the auditory portions of the inner ear during both sound pressure and particle motion. The design is clever and is novel in that it could image this structures during stimulation. The authors do a good job of indicating potential limitations with their procedures. The study makes a very nice overall contribution to the field.

Thank you very much for the positive feedback on our ms. 

A couple of items that need to be clarified or considered.

The fish are dead and tissue structure can break down very quickly, especially if high intensity lighting was needed and it generated heat. It would have been useful if there was postmortem examination to determine if there were any structure changes that could not be detected by imaging. For example, what was the effect of blood clotting?

We would like to refer to our statement made in the Material and methods section:

“To minimize detrimental effects of X-ray-induced tissue damage, experiments performed in the same orientation (e.g. fish seen in lateral view) and using the same species, but different individuals, alternately started with the in phase (0°) and the out of phase condition (180°; for an explanation of the two conditions see “setup design”). Hence, we started using the in phase condition and the highest studied sound pressure level (SPL; step 1, Table 1) in one specimen and using the out of phase condition at a lower SPL (e.g. step 4, Table 1) in another individual of the same species.”

In the discussion of the revised version of the ms, we included a short statement that future studies should compare tissue damage on the histological level by comparing control fish with specimens subjected to hard X-ray imaging.

The SPL were very high and not normally used in fish auditory studies. Was this due to the limitations in the imaging? I would expect to see large scale movements at these SPL that fish may never experience. How did this impact the studies and models?

Yes, this is exactly the point that we had to find a compromise between the acoustics and the constraints implied by the imaging procedure to obtain a meaningful signal-to-noise ratio. We had to go for rather high SPLs because the currently achievable spatial and temporal resolution is not high enough to visualize otoliths moving only a few nanometers as would be expected for lower SPLs (e.g. at the auditory threshold). We agree that this is an important aspect and we thus briefly address this issue in the discussion in the revised version of the ms.

I presume the test chamber is designed to get around the problems associated with auditory studies in small tanks. Can physiological studies eventually be done in these types of tanks?

We think that this is a very interesting aspect. It may be feasible to measure e.g. AEPs during sound stimulation while imaging the moving auditory structures, if it was possible to subject live animals to an acceptable X-ray dosage while still yielding a good signal-to-noise ratio.

I felt the discussion is far too speculative. The authors have a very nice data set and do not need to explain every variance that may be impacting the structure. I would suggest that the discussion focuses on what can definitely be determined and limit the number of speculative discussion.

As recommended, we shortened the discussion, especially with regard to potential applications of the setup.

I realize this is an on line journal and page limits are not a concern, however 18 figures seem a little excessive and it is hard for the reader to wade through and determine which ones are really need. I would suggest that several figures could be moved to supplemental files.

As suggested, we reduced the number of figures in the main text to 11 instead of 18 and moved 7 figures to the Supporting information.

Minor

Abstract The use of the term maximum even in quotations is confusing.

Corrected as suggested. We rephrased these sentences using “maximised” instead of “maximum”.

29 “inner” ears. Many times the authors use the term ear, when inner ear would be more accurate

Corrected as suggested. We now use “inner ears” instead of just “ears” throughout the manuscript.

40 I think it is safe to say they are no longer “assumed”

Corrected as suggested. Now it reads “…and play an important role…”

45-46. Sentences end and start with same words

We rephrased this sentence to avoid at least part of the repetition. It reads now as follows: “These structures….”

What is an improved hearing ability. This is usually termed increased sensitivity or greater frequency range

We rephrased the half-sentence to be more precise/specific: “….which are often correlated with a widened range of detectable frequencies (especially at frequencies > 700 Hz) and increased auditory sensitivity.”

51 … whole body in motion: It is unclear the effect of hard skeletal elements. Also, the motion is oscillolatory with no net movement gain and therefore motion is not best word? Vibration?

We rephrased this accordingly: “….results in a to and fro motion of the fish..” to make clear that it is not a net movement into a certain direction.

78 The pearlspot is not considered as well investigated as the goldfish

We agree that goldfish and E. canarensis are not equally well investigated with regard to the auditory structures and auditory abilities. We therefore rephrased this part as follows:

“Goldfish is well investigated with regard to inner ear and swim bladder morphology. For Etroplus, data on the gross morphology of the auditory structures are available.”

95 The saccule has plays a very important vestibular function and therefore does not “mainly” function in audition

We corrected the sentence accordingly:

“1) In otophysans such as goldfish, the saccule is assumed to play an important role in perceiving sound pressure.”

113 The cessation of opercular movement is not considered an endpoint of euthanasia. Many fish (including goldfish) can be easily revived after the cessation of opercular movement. There should have been an additional period following the cessation of movement to insure complete euthanasia

As we used a strong solution of the anesthetic (0.4% buffered MS-222), opercular movements ceased rather quickly, i.e. within 5 minutes. After that we waited further 10-15 minutes before further handling of the fish. In the revised version of the ms, we clarified this accordingly.

Methods

The image rate of 200 Hz for the 200 Hz seems relatively slow. Most video is run higher than stimulus. Please explain if this had an effect on resolving the movements

We would like to refer to a section in the original version of the ms addressing this issue: “From an imaging point of view, the “large” water body in the tank impairs a real-time observation of the moving auditory structures in the fish. At ID19, maximum image acquisition rates of up to 1 kHz (1,000 fps) are possible, which would be sufficient to record these motions in real time. However, such high frame rates are only possible for samples investigated in air [60,61]. It is thus important to note that our observations of moving otoliths and ancillary auditory structures are a proxy of the true motion.”

633 add reference

Corrected as suggested. We included the respective reference, i.e. Vetter et al. (2019) J Comp Physiol A.

Table 1: delete, add info to text

As recommended, we deleted Table 1 and included the information in the text.

---

## [Editor Report · Decision Letter 1]

4 Mar 2020

Auditory chain reaction: effects of sound pressure and particle motion on auditory structures in fishes

PONE-D-19-30473R1

Dear Dr. Schulz-Mirbach,

We are pleased to inform you that your manuscript has been judged scientifically suitable for publication and will be formally accepted for publication once it complies with all outstanding technical requirements.

With kind regards,

Dennis M. Higgs

Academic Editor

PLOS ONE
---

## [Editor Report · Acceptance letter]

16 Mar 2020

PONE-D-19-30473R1 

Auditory chain reaction: effects of sound pressure and particle motion on auditory structures in fishes 

Dear Dr. Schulz-Mirbach:

I am pleased to inform you that your manuscript has been deemed suitable for publication in PLOS ONE. Congratulations! Your manuscript is now with our production department. 

With kind regards,

on behalf of

Dr. Dennis M. Higgs 

Academic Editor

PLOS ONE